# Learning MDPs from Features:
# Predict-Then-Optimize for Sequential Decision Problems by Reinforcement Learning

**Kai Wang**
Harvard University
Cambridge, MA
kaiwang@g.harvard.edu

**Sanket Shah**
Harvard University
Cambridge, MA
sanketshah@g.harvard.edu

**Haipeng Chen**
Harvard University
Cambridge, MA
hpchen@seas.harvard.edu

**Andrew Perrault**[*]
The Ohio State University
Columbus, OH
perrault.17@osu.edu

**Finale Doshi-Velez**
Harvard University
Cambridge, MA
finale@seas.harvard.edu

**Milind Tambe**
Harvard University
Cambridge, MA
milind_tambe@harvard.edu

## Abstract

In the predict-then-optimize framework, the objective is to train a predictive model, mapping from environment features to parameters of an optimization problem, which maximizes decision quality when the optimization is subsequently solved. Recent work on decision-focused learning shows that embedding the optimization problem in the training pipeline can improve decision quality and help generalize better to unseen tasks compared to relying on an intermediate loss function for evaluating prediction quality. We study the predict-then-optimize framework in the context of *sequential* decision problems (formulated as MDPs) that are solved via reinforcement learning. In particular, we are given environment features and a set of trajectories from training MDPs, which we use to train a predictive model that generalizes to unseen test MDPs without trajectories. Two significant computational challenges arise in applying decision-focused learning to MDPs: (i) large state and action spaces make it infeasible for existing techniques to differentiate through MDP problems, and (ii) the high-dimensional policy space, as parameterized by a neural network, makes differentiating through a policy expensive. We resolve the first challenge by sampling provably unbiased derivatives to approximate and differentiate through optimality conditions, and the second challenge by using a low-rank approximation to the high-dimensional sample-based derivatives. We implement both Bellman–based and policy gradient–based decision-focused learning on three different MDP problems with missing parameters, and show that decision-focused learning performs better in generalization to unseen tasks.

## 1 Introduction

*Predict-then-optimize* [5, 9] is a framework for solving optimization problems with unknown parameters. Given such a problem, we first train a predictive model to predict the missing parameters from problem features. Our objective is to maximize the resulting decision quality when the optimization problem is subsequently solved with the predicted parameters [25, 28]. Recent work on the *decision-focused learning* approach [7, 36] embeds the optimization problem [1, 3, 4] into the training pipeline and trains the predictive model end-to-end to optimize the final decision quality. Compared with a

---

[*]This work was done while the author was at Harvard University.

35th Conference on Neural Information Processing Systems (NeurIPS 2021).

more traditional "two-stage" approach which maximizes the predictive accuracy of the model (rather than the final decision quality), the decision-focused learning approach can achieve a higher solution quality and generalize better to unseen tasks.

This paper studies the predict-then-optimize framework in *sequential* decision problems, formulated as Markov decision processes (MDPs), with unknown parameters. In particular, at training time, we are given trajectories and environment features from "training MDPs." Our goal is to learn a predictive model which maps from environment features to missing parameters based on these trajectories that generalizes to unseen test MDPs that have features, but not trajectories. The resulting "predicted" training and test MDPs are solved using deep reinforcement learning (RL) algorithms, yielding policies that are then evaluated by offline off-policy evaluation (OPE) as shown in Figure 1. This fully offline setting is motivated by real-world applications such as wildlife conservation and tuberculosis treatment where no simulator is available. However, such domains offer past ranger patrol trajectories and environmental features of individual locations from conservation parks for generalization to other unpatrolled areas. These settings differ from those considered in transfer-RL [21, 24, 29, 31] and meta-RL [8, 10, 33, 35, 40] because we generalize across different MDPs by explicitly predicting the mapping function from features to missing MDPs parameters, while transfer/meta RL achieve generalization by learning hidden representation of different MDPs implicitly with trajectories.

The main contribution of this paper is to extend the decision-focused learning approach to MDPs with unknown parameters, embedding the MDP problems in the predictive model training pipeline. To perform this embedding, we study two common types of optimality conditions in MDPs: a Bellman-based approach where mean-squared Bellman error is minimized, and a policy gradient-based approach where the expected cumulative reward is maximized. We convert these optimality conditions into their corresponding Karush–Kuhn–Tucker (KKT) conditions, where we can backpropagate through the embedding by differentiating through the KKT conditions. However, existing techniques from decision-focused learning and differentiating through KKT conditions do not directly apply as the size of the KKT conditions of sequential decision problems grow linearly in the number of states and actions, which are often combinatorial or continuous and thus become intractable.

We identify and resolve two computational challenges in applying decision-focused learning to MDPs, that come up in both optimality conditions: (i) the large state and action spaces involved in the optimization reformulation make differentiating through the optimality conditions intractable and (ii) the high-dimensional policy space in MDPs, as parameterized by a neural network, makes differentiating through a policy expensive. To resolve the first challenge, we propose to sample an estimate of the first-order and second-order derivatives to approximate the optimality and KKT conditions. We prove such a sampling approach is unbiased for both types of optimality conditions. Thus, we can differentiate through the approximate KKT conditions formed by sample-based derivatives. Nonetheless, the second challenge still applies—the sampled KKT conditions are expensive to differentiate through due to the dimensionality of the policy space when model-free deep RL methods are used. Therefore, we propose to use a low-rank approximation to further approximate the sample-based second-order derivatives. This low-rank approximation reduces both the computation cost and the memory usage of differentiating through KKT conditions.

We empirically test our decision-focused algorithms on three settings: a grid world with unknown rewards, and snare-finding and Tuberculosis treatment problems where transition probabilities are unknown. Decision-focused learning achieves better OPE performance in unseen test MDPs than two-stage approach, and our low-rank approximations significantly scale-up decision-focused learning.

## Related Work

**Differentiable optimization**     Amos et al. [2] propose using a quadratic program as a differentiable layer and embedding it into deep learning pipeline, and Agrawal et al. [1] extend their work to convex programs. Decision-focused learning [7, 36] focuses on the predict-then-optimize [5, 9] framework by embedding an optimization layer into training pipeline, where the optimization layers can be convex [7], linear [22, 36], and non-convex [26, 34]. Unfortunately, these techniques are of limited utility for sequential decision problems because their formulations grow linearly in the number of states and actions and thus differentiating through them quickly becomes infeasible. Amos et al. [3] avoid this issue by studying model-predictive control but limited to quadratic-form actions, reducing the dimensionality. Karkus et al. [16] differentiate through an algorithm by unrolling and relaxing

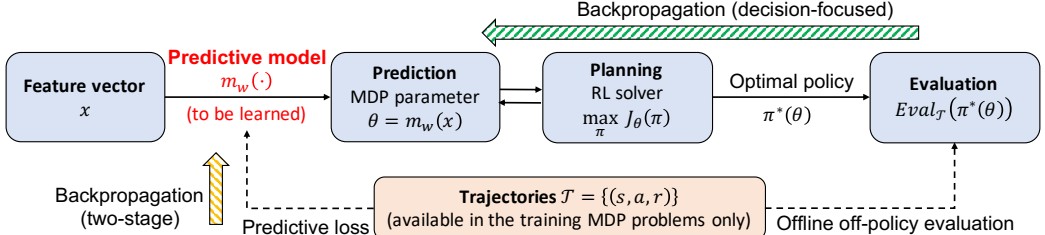

Figure 1: We consider learning a predictive model to map from features to unknown MDP parameters and obtaining a policy by solving the predicted MDP with RL. Two-stage learning learns the predictive model by minimizing a predictive loss function, whereas decision-focused learning is trained end-to-end to maximize the final off-policy evaluation performance.

all the strict operators by soft operators. Futoma et al. [11] deal with large optimality conditions by differentiating through the last step of the value-iteration algorithm only. Instead, our approach does not rely on any MDP solver structure. We combine sampling and a low-rank approximation to form an unbiased estimate of the optimality conditions to differentiate through, and show that the approach of Futoma et al. [11] is included in ours as a special case.

**Predict-then-optimize and offline reinforcement learning**   The idea of planning under a predicted MDP arises in model-based RL as *certainty equivalence* [19]. It has been extended to offline settings [17, 39], who learn a pessimistic MDP before solving for the policy. Our setting differs because of the presence of features and train-test split—our test MDPs are completely fresh *without any trajectories*. Our setting also resembles meta RL (e.g., [8, 10, 33, 35, 40]) and transfer RL (e.g., [21, 24, 29, 31].) Meta RL focuses on training a "meta policy" for a distribution of tasks (MDPs), leveraging trajectories for each. Transfer RL works by extracting transferable knowledge from source MDPs to target MDPs using trajectories. In contrast to these two paradigms, ours explicitly trains a predictive model (which maps problem features to missing MDP parameters) to generalize knowledge learned from the training set to the testing set using *problem features, not trajectories*.

## 2   Problem Statement

In this paper, we consider learning a predictive model to infer the missing parameters in a sequential decision-making task (formulated as MDPs) using the predict-then-optimize framework. Each MDP is defined by a tuple $(\mathcal{S}, \boldsymbol{s}_0, \mathcal{A}, T, R)$ with an initial state $\boldsymbol{s}_0$, a possibly infinite state space $\mathcal{S}$ and possibly infinite action space $\mathcal{A}$. We assume some parameters are missing in each MDP, which could be any portion of the transition function $T$ and the reward function $R$. We denote the missing parameters vector by $\theta^*$. Additionally, we assume there are problem features $x$ associated with each MDP, where $(\theta^*, x)$ is correlated and drawn from the same unknown, but fixed, distribution[2].

We are given a set of training MDPs and a set of test MDPs, each with missing parameters $\theta^*$ and features $x$. Each training MDP is accompanied by a set of trajectories $\mathcal{T}$ performed by a behavior policy $\pi_{\text{beh}}$, consisting of a sequence of states, actions and rewards. In the test MDPs, trajectories from the behavior policy are hidden at test time. These testing MDPs are considered fresh instances that we have to generate a policy without using any trajectories. Thus, at training time, we learn a predictive model $m_w$ to map from features to missing parameters; at test time, we apply the same model to make predictions and plan accordingly without using trajectories.

Formally, our goal is to learn a predictive model $m_w$ to predict the missing parameters $\theta = m_w(x)$. The predicted parameters are used to solve the test MDPs, yielding the policy $\pi^*(m_w(x))$ where our offline off-policy evaluation (OPE) metric is maximized. This framework is illustrated in Figure 1.

---

[2]Examples of the missing parameters $\theta^*$ include the poaching risk of different locations in wildlife conservation and the transition probability of patients' healthiness in healthcare problems, where the corresponding problem features are terrain features of different locations and the characteristics of different patients that are correlated to the missing parameters, respectively. These correlated features allow us to predict the missing parameters even if we do not have any trajectories of the MDP.

**Offline off-policy evaluation**   We evaluate a policy $\pi$ in a fully offline setting with trajectories $\mathcal{T} = \{\tau_i\}, \tau_i = (s_{i1}, a_{i1}, r_{i1}, \cdots, s_{ih}, a_{ih}, r_{ih})$ generated from the MDP using behavior policy $\pi_{\text{beh}}$. We use an OPE metric used by Futoma et al. [11] — we evaluate a policy $\pi$ and trajectories $\mathcal{T}$ as:

$$\text{Eval}_{\mathcal{T}}(\pi) := V^{\text{CWPDIS}}(\pi) - \lambda_{\text{ESS}}/\sqrt{\text{ESS}(\pi)} \tag{1}$$

where $V^{\text{CWPDIS}}(\pi) := \sum_{t=1}^{h} \gamma^t \frac{\sum_i r_{it} \rho_{it}(\pi)}{\sum_i \rho_{it}(\pi)}$ and $\text{ESS}(\pi) := \sum_{t=1}^{h} \frac{(\sum_i \rho_{it})^2}{\sum_i \rho_{it}^2}$, and $\rho_{it}(\pi)$ is the ratio of the proposed policy and the behavior policy likelihoods up to time t: $\rho_{it}(\pi) := \prod_{t'=1}^{t} \frac{\pi(a_{it'}|s_{it'})}{\pi_{\text{beh}}(a_{it'}|s_{it'})}$.

**Optimization formulation**   Given a set of training features and trajectories $D_{\text{train}} = \{(x_i, \mathcal{T}_i)\}_{i \in I_{\text{train}}}$, our goal is to learn a predictive model $m_w$ to optimize the training performance:

$$\max_w \quad \mathbb{E}_{(x, \mathcal{T}) \in D_{\text{train}}} \left[ \text{Eval}_{\mathcal{T}}(\pi^*(m_w(x))) \right] \tag{2}$$

The testing performance is evaluated on the unseen test set $D_{\text{test}} = \{(x_i, \mathcal{T}_i)\}_{i \in I_{\text{test}}}$ with trajectories hidden from training, and only used for evaluation: $\mathbb{E}_{(x, \mathcal{T}) \in D_{\text{test}}} \left[ \text{Eval}_{\mathcal{T}}(\pi^*(m_w(x))) \right]$.

## 3   Two-stage Learning

To learn the predictive model $m_w$ from trajectories, the standard approach is to minimize an expected predictive loss, which is defined by comparing the prediction $\theta = m_w(x)$ with the trajectories $\mathcal{T}$:

$$\min_w \quad \mathbb{E}_{(x, \mathcal{T}) \sim \mathcal{D}_{\text{train}}} \mathcal{L}(\theta, \mathcal{T}) \qquad \text{where} \quad \mathcal{L}(\theta, \mathcal{T}) = \mathbb{E}_{\tau \sim \mathcal{T}} \ell_\theta(\tau), \quad \theta = m_w(x) \tag{3}$$

For example, when the rewards are missing, the loss could be the squared error between the predicted rewards and the actual rewards we see in the trajectories for each individual step. When the transition probabilities are missing, the loss could be defined as the negative log-likelihood of seeing the trajectories in the training set.

In the first stage, to train the predictive model, we run gradient descent to minimize the loss function defined in Equation (3) and make prediction on the model parameter $\theta = m_w(x)$ of each problem. In the second stage, we solve each MDP problem with the predicted parameter $\theta$ using an RL algorithm to generate the optimal policy $\pi^*(\theta)$. However, predictive loss and the final evaluation metric are commonly misaligned especially in deep learning problems with imbalanced data [6, 14, 15, 20]. This motivates us to learn the predictive model end-to-end and therefore avoid the misalignment.

## 4   Decision-focused Learning

In this section, we present our main contribution, decision-focused learning in sequential decision problems, as illustrated in Figure 1. Decision-focused learning integrates an MDP problem into the training pipeline to directly optimize the final performance. Instead of relying on a predictive loss to train the predictive model $m_w$, we can directly optimize the objective in Equation (2) by running end-to-end gradient descent to update the predictive model $m_w$:

$$\frac{d \, \text{Eval}(\pi^*)}{dw} = \frac{d \, \text{Eval}(\pi^*)}{d\pi^*} \frac{d\pi^*}{d\theta} \frac{d\theta}{dw} \tag{4}$$

We assume the policy $\pi^*(\theta)$ is either stochastic and smooth with respect to the change in the parameter $\theta$, which is common in settings with continuous state or action spaces, or that an appropriate regularization term [12, 13] is used to improve the smoothness of the policy. More discussions about the smoothness can be found in Appendix B.1.

This gradient computation requires us to back-propagate from the final evaluation through the MDP layer to the predictive model $m_w$ that we want to update. The major challenge in Equation (4) is to compute $\frac{d\pi^*}{d\theta}$, which involves differentiating through an MDP layer solved by an RL algorithm. In the following section, we first discuss two different optimality conditions in MDPs, which are later used to convert into KKT conditions and differentiate through to compute $\frac{d\pi^*}{d\theta}$. We then discuss two computational challenges associated with the derivative computation.

## 4.1 Optimality Conditions in MDPs

When the predicted model parameter $\theta = m_w(x)$ is given, the MDP can be solved by any RL algorithm to get an optimal policy $\pi^*$. Here we discuss two common optimality conditions in MDPs, differing by the use of policy gradient or Bellman equation:

**Definition 1** (Policy gradient-based optimality condition). *Defining $J_\theta(\pi)$ to be the expected cumulative reward under policy $\pi$, the optimality condition of the optimal policy $\pi^*$ is:*

$$\pi^* = \arg\max_\pi J_\theta(\pi) \qquad where \quad J_\theta(\pi) := \mathbb{E}_{\tau \sim \pi, \theta} \, G_\theta(\tau) \qquad (5)$$

*where $G_\theta(\tau)$ is the discounted value of trajectory $\tau$ given parameter $\theta$, and the expectation is taken over the trajectories following the optimal policy and transition probability (as part of $\theta$).*

**Definition 2** (Bellman-based optimality condition). *Defining $J_\theta(\pi)$ to be the mean-squared Bellman error[3] under policy $\pi$, the optimality condition of the optimal policy $\pi^*$ (valuation function) is:*

$$\pi^* = \arg\min_\pi J_\theta(\pi) \qquad where \quad J_\theta(\pi) := \mathbb{E}_{\tau \sim \pi, \theta} \, \frac{1}{2} \delta_\theta^2(\tau, \pi) \qquad (6)$$

*where $\delta_\theta(\tau, \pi) = \sum_{(s,a,s') \in \tau} Q_\pi(s, a) - R_\theta(s, a) - \gamma \, \mathbb{E}_{a' \sim \pi} Q_\pi(s', a')$ is the total Bellman error of a trajectory $\tau$, and each individual term $\delta_\theta(\tau, \pi)$ can depend on the parameter $\theta$ because the Bellman error depends on the immediate reward $R_\theta$, which can be a part of the MDP parameter $\theta$. The expectation in Equation (6) is taken over all the trajectories generated from policy $\pi$ and transition probability (as part of $\theta$).*

## 4.2 Backpropagating Through Optimality and KKT Conditions

To compute the derivative of the optimal policy $\pi^*(\theta)$ in an MDP with respect to the MDP parameter $\theta$, we differentiate through the KKT conditions of the corresponding optimality conditions:

**Definition 3** (KKT Conditions). *Given objective $J_\theta(\pi)$ in an MDP problem, since the policy parameters are unconstrained, the necessary KKT conditions can be written as: $\nabla_\pi J_\theta(\pi^*) = 0$.*

In particular, computing the total derivative of KKT conditions gives:

$$0 = \frac{d}{d\theta} \nabla_\pi J_\theta(\pi^*) = \frac{\partial}{\partial\theta} \nabla_\pi J_\theta(\pi^*) + \frac{\partial}{\partial\pi} \nabla_\pi J_\theta(\pi^*) \frac{d\pi^*}{d\theta} = \nabla_{\theta\pi} J_\theta(\pi^*) + \nabla_\pi^2 J_\theta(\pi^*) \frac{d\pi^*}{d\theta}$$

$$\implies \frac{d\pi^*}{d\theta} = -(\nabla_\pi^2 J_\theta(\pi^*))^{-1} \nabla_{\theta\pi}^2 J_\theta(\pi^*) \qquad (7)$$

We can use Equation (7) to replace the term $\frac{d\pi^*}{d\theta}$ in Equation (4) to compute the full gradient to back-propagate from the final evaluation to the predictive model parameterized by $w$:

$$\frac{d \, \text{Eval}(\pi^*)}{dw} = -\frac{d \, \text{Eval}(\pi^*)}{d\pi^*} (\nabla_\pi^2 J_\theta(\pi^*))^{-1} \nabla_{\theta\pi}^2 J_\theta(\pi^*) \frac{d\theta}{dw} \qquad (8)$$

## 4.3 Computational Challenges in Backward Pass

Unfortunately, although we can write down and differentiate through the KKT conditions analytically, we cannot explicitly compute the second-order derivatives $\nabla_\pi^2 J_\theta(\pi^*)$ and $\nabla_{\theta\pi}^2 J_\theta(\pi^*)$ in Equation (8) due to the following two challenges:

**Large state and action spaces involved in optimality conditions** The objectives $J_\theta(\pi^*)$ in Definition 1 and Definition 2 involve an expectation over all possible trajectories, which is essentially an integral and is intractable when either the state or action space is continuous. This prevents us from explicitly verifying optimality and writing down the two derivatives $\nabla_\pi^2 J_\theta(\pi^*)$ and $\nabla_{\theta\pi}^2 J_\theta(\pi^*)$.

**High-dimensional policy space parameterized by neural networks** In MDPs solved by model-free deep RL algorithms, the policy space $\pi \in \Pi$ is often parameterized by a neural network, which has a significantly larger number of variables than standard optimization problems. This large dimensionality makes the second-order derivative $\nabla_\pi^2 J_\theta(\pi^*) \in \mathbb{R}^{\dim(\pi) \times \dim(\pi)}$ intractable to compute, store, or invert.

---

[3]We use the same notation $J$ to denote both the expected cumulative reward and the expected Bellman error to simplify the later analysis of decision-focused learning.

# 5 Sampling Unbiased Derivative Estimates

In both policy gradient–based and Bellman–based optimality conditions, the objective is implicitly given by an expectation over all possible trajectories, which could be infinitely large when either state or action space is continuous. This same issue arises when expressing such an MDP as a linear program — there are infinitely many constraints, making it intractable to differentiate through.

Inspired by the policy gradient theorem, although we cannot compute the exact gradient of the objective, we can sample a set of trajectories $\tau = \{s_1, a_1, r_1, \ldots, s_h, a_h, r_h\}$ from policy $\pi$ and model parameter $\theta$ with finite time horizon $h$. Denoting $p_\theta(\tau, \pi)$ to be the likelihood of seeing trajectory $\tau$, we can compute an unbiased derivative estimate for both optimality conditions:

**Theorem 1** (Policy gradient-based unbiased derivative estimate). *We follow the notation of Definition 1 and define $\Phi_\theta(\tau, \pi) = \sum_{i=1}^{h} \sum_{j=i}^{h} \gamma^j R_\theta(s_j, a_j) \log \pi(a_i | s_i)$. We have:*

$$\nabla_\pi J_\theta(\pi) = \mathbb{E}_{\tau \sim \pi, \theta} \left[ \nabla_\pi \Phi_\theta(\tau, \pi) \right] \implies \begin{aligned} \nabla_\pi^2 J_\theta(\pi) &= \mathbb{E}_{\tau \sim \pi, \theta} \left[ \nabla_\pi \Phi_\theta \cdot \nabla_\pi \log p_\theta^\top + \nabla_\pi^2 \Phi_\theta \right] \\ \nabla_{\theta\pi}^2 J_\theta(\pi) &= \mathbb{E}_{\tau \sim \pi, \theta} \left[ \nabla_\pi \Phi_\theta \cdot \nabla_\theta \log p_\theta^\top + \nabla_{\theta\pi}^2 \Phi_\theta \right] \end{aligned} \quad (9)$$

**Theorem 2** (Bellman-based unbiased derivative estimate). *We follow the notation in Definition 2 to define $J_\theta(\pi) = \frac{1}{2} \mathbb{E}_{\tau \sim \pi, \theta} \left[ \delta_\theta^2(\tau, \pi) \right]$. We have:*

$$\nabla_\pi J_\theta(\pi) = \mathbb{E}_{\tau \sim \pi, \theta} \left[ \delta \nabla_\pi \delta + \frac{1}{2} \delta^2 \nabla_\pi \log p_\theta \right] \implies \nabla_\pi^2 J_\theta(\pi) = \mathbb{E}_{\tau \sim \pi, \theta} \left[ \nabla_\pi \delta \nabla_\pi \delta^\top + O(\delta) \right]$$

$$\nabla_{\theta\pi}^2 J_\theta(\pi) = \mathbb{E}_{\tau \sim \pi, \theta} \left[ \nabla_\pi \delta \nabla_\theta \delta^\top + \left( \nabla_\pi \delta \nabla_\theta \log p_\theta^\top + \nabla_\pi \log p_\theta \nabla_\theta \delta^\top + \nabla_{\theta\pi}^2 \delta \right) \delta + O(\delta^2) \right] \quad (10)$$

For the latter, we apply the fact that at the near-optimal policy, the Bellman error is close to $0$ and thus each individual component $\delta(\tau, \pi)$ is small to simplify the analysis. Refer to the appendix for the full derivations of Equations (9) and (10).

Equations (9) and (10) provide a sampling approach to compute the second-order derivatives, avoiding computing an expectation over the large trajectory space. We can use the optimal policy derived in the forward pass and the predicted parameters $\theta$ to run multiple simulations to collect a set of trajectories. These trajectories from predicted parameters can be used to compute each individual derivative in Equations (9) and (10).

# 6 Resolving High-dimensional Derivatives by Low-rank Approximation

Section 5 provides sampling approaches to compute an unbiased estimate of second-order derivatives. However, since the dimensionality of the policy space $\dim(\pi)$ is often large, we cannot explicitly expand and invert $\nabla_\pi^2 J_\theta(\pi^*)$ to compute $\nabla_\pi^2 J_\theta(\pi^*)^{-1} \nabla_{\theta\pi}^2 J_\theta(\pi^*)$, which is an inevitable step toward computing the full gradient of decision-focused learning in Equation (8). In this section, we discuss various ways to approximate $\nabla_\pi^2 J_\theta(\pi^*)$ and how we use low-rank approximation and Woodbury matrix identity [37] to efficiently invert the sampled Hessian without expanding the matrices. Let $n := \dim(\pi)$ and $k \ll n$ to be the number of trajectories we sample to compute the derivatives.

## 6.1 Full Hessian Computation

In Equations (9) and (10), we can apply auto-differentiation tools to compute all individual derivatives in the expectation. However, this works only when the dimensionality of the policy space $\pi \in \Pi$ is small because the full expressions in Equations (9) and (10) involve computing second-order derivatives , e.g., $\nabla_\pi^2 \Phi_\theta$ in Equation (10), which is still challenging to compute and store when the matrix size $n \times n$ is large. The computation cost is $O(n^2 k) + O(n^\omega)$ dominated by computing all the Hessian matrices and the matrix inversion with $2 \leq \omega \leq 2.373$ the complexity order of matrix inversion.

## 6.2 Approximating Hessian by Constant Identity Matrix

One naive way to approximate the Hessian $\nabla_\pi^2 J_\theta(\pi^*)$ is to simply use a constant identity matrix $cI$. We choose $c < 0$ for the policy gradient–based optimality in Definition 1 because the optimization

---

**Algorithm 1:** Decision-focused Learning for MDP Problems with Missing Parameters

---

**1 Parameter:** Training set $\mathcal{D}_{\text{train}} = \{(x_i, \mathcal{T}_i)\}_{i \in I_{\text{train}}}$, learning rate $\alpha$, regularization $\lambda = 0.1$

**2 Initialization:** Initialize predictive model $m_w : \mathcal{X} \to \Theta$ parameterized by $w$

**3 for** *epoch* $\in \{1, 2, \dots\}$, *each training instance* $(x, \mathcal{T}) \in \mathcal{D}_{train}$ **do**

**4**      **Forward:** Make prediction $\theta = m_w(x)$. Compute two-stage loss $\mathcal{L}(\theta, \mathcal{T})$. Run model-free RL to get an optimal policy $\pi^*(\theta)$ on MDP problem using parameter $\theta$.

**5**      **Backward:** Sample a set of trajectories under $\theta, \pi^*$ to compute $\nabla^2_\pi J_\theta(\pi^*), \nabla^2_{\theta\pi} J_\theta(\pi^*)$

**6**      **Gradient step:** Set $\Delta w = \frac{d \operatorname{Eval}_{\mathcal{T}}(\pi^*)}{dw} - \lambda \frac{d\mathcal{L}(\theta, \mathcal{T})}{dw}$ by Equation (8) with predictive loss $\mathcal{L}$ as regularization. Run gradient ascent to update model: $w \leftarrow w + \alpha \Delta w$

**7 Return:** Predictive model $m_w$.

---

problem is a maximization problem and thus is locally concave at the optimal solution, whose Hessian is negative semi-definite. Similarly, we choose $c > 0$ for the Bellman–based optimality in Definition 2. This approach is fast, easily invertible. Moreover, in the Bellman version, Equation (8) is equivalent to the idea of differentiating through the final gradient of Bellman error as proposed by Futoma et al. [11][4]. However, this approach ignores the information provided by the Hessian term, which can often lead to instability as we later show in the experiments. In this case, the computation complexity is dominated by computing $\nabla^2_{\theta\pi} J_\theta(\pi^*)$, which requires $O(nk)$.

### 6.3 Low-rank Hessian Approximation and Application of Woodbury Matrix Identity

A compromise between the full Hessian and using a constant matrix is approximating the second-order derivative terms in Equations (9) and (10) by constant identity matrices, while computing the first-order derivative terms with auto-differentiation. Specifically, given a set of $k$ sampled trajectories $\{\tau_1, \tau_2, \cdots, \tau_k\}$, Equations (9) and (10) can be written and approximated in the following form:

$$\nabla^2_\pi J_\theta(\pi) \approx \frac{1}{k} \sum_{i=1}^k \left( u_i v_i^\top + H_i \right) \approx \frac{1}{k} \sum_{i=1}^k \left( u_i v_i^\top + cI \right) = UV^\top + cI \qquad (11)$$

where $U = [u_1, u_2, \cdots, u_k]/\sqrt{k} \in \mathbb{R}^{n \times k}, V = [v_1, v_2, \cdots, v_k]/\sqrt{k} \in \mathbb{R}^{n \times k}$ and $u_i, v_i \in \mathbb{R}^n$ correspond to the first-order derivatives in Equations (9) and (10), and $H_i$ corresponds to the remaining terms that involve second-order derivatives. We use a constant identity matrix to approximate $H_i$, while explicitly computing the remaining parts to increase accuracy.

However, we still cannot explicitly expand $UV^\top \in \mathbb{R}^{n \times n}$ since the dimensionality is too large. Therefore, we apply Woodbury matrix identity [37] to invert Equation (11):

$$(\nabla^2_\pi J_\theta(\pi))^{-1} \approx (UV^\top + cI)^{-1} = \frac{1}{c}I - \frac{1}{c}U(cI - V^\top U)^{-1}V^\top \qquad (12)$$

where $V^\top U \in \mathbb{R}^{k \times k}$ can be efficiently computed with much smaller $k \ll n$. When we compute the full gradient for decision-focused learning in Equation (8), we can then apply matrix-vector multiplication without expanding the full high-dimensional matrix, which results in a computation cost of $O(nk + k^\omega)$ that is much smaller than the full computation cost $O(n^2k + n^\omega)$.

The full algorithm for decision-focused learning in MDPs is illustrated in Algorithm 1.

## 7 Example MDP Problems with Missing Parameters

**Gridworld with unknown cliffs** We consider a Gridworld environment with a set of training and test MDPs. Each MDP is a $5 \times 5$ grid with a start state located at the bottom left corner and a safe state with reward drawn from $\mathcal{N}(5, 1)$ located at the top right corner. Each intermediate state has a reward associated with it, where most of them give the agent a reward drawn from $\mathcal{N}(0, 1)$ but 20% of the them are cliffs and give $\mathcal{N}(-10, 1)$ penalty to the agent. The agent has no prior information

---

[4]The gradient of Bellman error can be written as $\nabla_\pi J_\theta(\pi^*)$ where the policy $\pi$ is the parameters of the value function approximator and $J$ is defined as the expected Bellman error. The derivative of the final gradient can be written as $\nabla_w(\nabla_\pi J_\theta(\pi^*)) = \nabla^2_{\theta\pi} J_\theta(\pi^*)\frac{d\theta}{dw}$ by chain rule, which matches the last three terms in Equation 8 when the Hessian is approximated by an identity matrix.

Table 1: OPE performances of different methods on the test MDPs averaged over 30 independent runs. Decision-focused learning methods consistently outperform two-stage approach, with some exception using identity matrix based Hessian approximation which may lead to high gradient variance.

| Trajectories | Gridworld | | Snare | | Tuberculosis | |
|---|---|---|---|---|---|---|
| | Random | Near-optimal | Random | Near-optimal | Random | Near-optimal |
| TS | $-12.0\pm1.3$ | $4.2\pm0.8$ | $0.8\pm0.3$ | $3.7\pm0.3$ | $35.8\pm1.5$ | $38.7\pm1.6$ |
| PG-Id | $-11.7\pm1.2$ | $5.7\pm0.8$ | $-0.1\pm0.3$ | $3.6\pm0.3$ | $38.4\pm1.5$ | $40.7\pm1.7$ |
| Bellman-Id | $-9.6\pm1.4$ | $4.6\pm0.7$ | $0.7\pm0.4$ | $3.6\pm0.3$ | $39.1\pm1.7$ | $40.8\pm1.7$ |
| PG-W | $-11.2\pm1.2$ | $5.5\pm0.8$ | $1.2\pm0.4$ | $4.8\pm0.3$ | $38.4\pm1.5$ | $40.8\pm1.7$ |
| Bellman-W | $-11.3\pm1.4$ | $4.8\pm0.8$ | $1.5\pm0.4$ | $4.3\pm0.3$ | $38.6\pm1.6$ | $41.1\pm1.7$ |

about the reward of each grid cell (i.e., the reward functions of the MDPs are unknown), but has a feature vector per grid cell correlated to the reward, and a set of historical trajectories from the training MDPs. The agent learns a predictive model to map from the features of a grid cell to its missing reward information, and the resulting MDP is used to plan. Since the state and action spaces are both finite, we use tabular value-iteration [30] to solve the MDPs.

**Partially observable snare-finding problems with missing transition function** We consider a set of *synthetic* conservation parks, each with 20 sites, that are vulnerable to poaching activities. Each site in a conservation park starts from a *safe* state and has an unknown associated probability that a poacher places a snare at each time step. Motivated by [38], we assume a ranger who can visit one site per time step and observes whether a snare is present. If a snare is present, the ranger removes it and receives reward 1. Otherwise, the ranger receives reward of $-1$. The snare can stay in the site if the ranger does not remove it, which makes the snare-finding problem a sequential problem rather than a multi-armed bandit problem. As the ranger receives no information about the sites that they do not visit, the MDP belief state is the ranger's belief about whether a snare is present. The ranger uses the features of each site and historical trajectories to learn a predictive model of the missing transition probability of a snare being placed. Since the belief state is continuous and the action space is discrete, given a predictive model of the missing transition probability, the agent uses double deep Q-learning (DDQN) [32] to solve the predicted MDPs.

**Partially observable patient treatment problems with missing transition probability** We consider a version of the Tuberculosis Adherence Problem explored in [23]. Given that the treatment for tuberculosis requires patients to take medications for an extended period of time, one way to improve patient adherence is Directly Observed Therapy, in which a healthcare worker routinely checks in on the patient to ensure that they are taking their medications. In our problem, we consider 5 *synthetic* patients who have to take medication for 30 days. Each day, a healthcare worker chooses one patient to intervene on. They observe whether that patient is adhering or not, and improve the patient's likelihood of adhering on that day, where we use the number of adherent patients as the reward to the healthcare worker. Whether a patient actually adheres or not is determined by a transition matrix that is randomly drawn from a fixed distribution inspired by [18]. The aim of the prediction stage is to use the features associated with each patient, e.g., patient characteristics, to predict the missing transition matrices. The aim of the RL stage is then to create an intervention strategy for the healthcare worker such that the sum of patient adherence over the 30-day period is maximised. Due to partial observability, we convert the problem to its continuous belief state equivalence and solve it using DDQN.

Please refer to Appendix C for more details about problem setup in all three domains.

# 8 Experimental Results and Discussion

In our experiments, we compare two-stage learning (**TS**) with different versions of decision-focused learning (**DF**) using two different optimality conditions, policy gradient (**PG**) and Bellman equation-based (**Bellman**), and two different Hessian approximations (**Identity**, **Woodbury**) defined in Section 6. Computing the **full** Hessian (as in Section 6.1) is computationally intractable. Across all three examples, we use 7 training MDPs, 1 validation MDP, and 2 test MDPs, each with 100 trajectories.

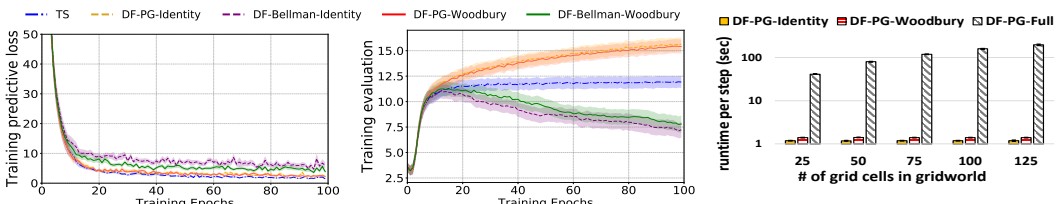

(a) Predictive loss in training MDPs    (b) Performance in training MDPs    (c) Backpropagation runtime per gradient step per instance

Figure 2: Learning curves of Gridworld problem with near-optimal trajectories. Two-stage minimizes the predictive loss in Figure 2(a), but this does not lead to good training performance in Figure 2(b). Figure 3(c) shows the backpropagation runtime per gradient step per instance of three Hessian approximations, which becomes intractable when trained for multiple instances and multiple epochs.

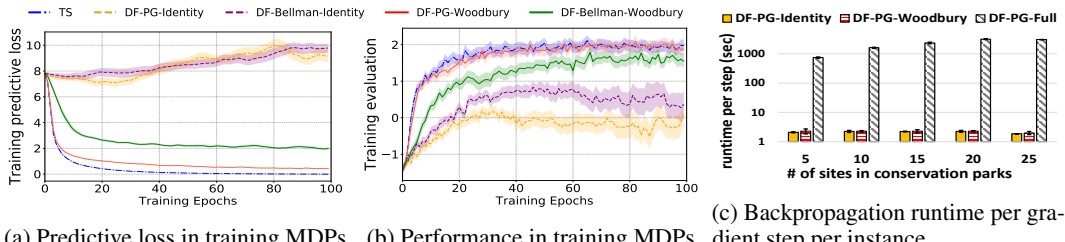

(a) Predictive loss in training MDPs    (b) Performance in training MDPs    (c) Backpropagation runtime per gradient step per instance

Figure 3: Learning curves of snare finding problems with random trajectories. Two-stage achieves both low predictive loss in Figure 3(a) and high training OPE in Figure 3(b), but the test performance is poor in Table 1. Figure 3(c) plots the backpropagation runtime per gradient step per instance.

The predictive model is trained on the training MDP trajectories for 100 epochs. Performance is evaluated under the Off-Policy Evaluation (OPE) metric of Equation (1) with respect to the withheld test trajectories. In the following, we will discuss *how* DF variants work compared with TS methods, and explore *why* some methods are better. We use two different trajectories, **random** and **near-optimal**, in the training MDP to model different imbalanced information given to train the predictive model. The results are shown in Table 1.

**Decision-focused learning with the Woodbury matrix identity outperforms two-stage learning**
Table 1 summarizes the average OPE performance on the test MDPs. We can see that in all of the three problem settings, the best performances are all achieved by decision-focused learning. However, when Hessian approximation is not sufficiently accurate, decision-focused learning can sometimes perform even worse than two-stage (e.g., PG-Id and Bellman-Id in the snare problem). In contrast, decision-focused methods using a more accurate low-rank approximation and Woodbury matrix identity (i.e., PG-W and Bellman-W), as discussed in Section 6.3, dominate two-stage performance in the test MDPs across all settings.

**Low predictive loss does not imply a winning policy**  In Figures 2(a), 3(a), we plot the predictive loss curve in the training MDPs over different training epochs of Gridworld and snare problems. In particular, two-stage approach is trained to minimize such loss, but fails to win in Table 1. Indeed, low predictive loss on the training MDPs does not always imply a high off-policy evaluation on the training MDPs in Figure 2(b) due to the misalignment of predictive accuracy and decision quality, which is consistent with other studies in mismatch of predictive loss and evaluation metric [6, 14, 15, 20].

**Comparison between different Hessian approximations**  In Table 1, we notice that more inaccurate Hessian approximation (identity) does not always lead to poorer performance. We hypothesize that this is due to the non-convex off-policy evaluation objective that we are optimizing, where higher variance might sometimes help escape local optimum more easily. The identity approximation is more unstable across different tasks and different trajectories given. In Table 1, the performance of Bellman-Identity and PG-Identity sometimes lead to wins over two stage and sometimes losses.

**Comparison between policy gradient and Bellman-based decision-focused learning**  We observe that the Bellman-based decision-focused approach consistently outperforms the policy gradient-based approach when the trajectories are random, while the policy gradient-based decision-focused approach mostly achieves better performance when near-optimal trajectories are present. We hypothesize that this is due to the different objectives considered by different optimality conditions. The Bellman error aims to accurately cover *all* the value functions, which works better on random trajectories; the policy gradient aims to maximize the expected cumulative reward along the *optimal policy only*, which works better with near-optimal trajectories that have better coverage in the optimal regions.

**Computation cost**  Lastly, Figures 2(c) and 3(c) show the backpropagation runtime of the policy-gradient based optimality condition per gradient step per training instance across different Hessian approximations and different problem sizes in the gridworld and snare finding problems. To train the model, we run 100 epochs for every MDP in the training set, which immediately makes the full Hessian computation intractable as it would take more than a day to complete.

Analytically, let $n$ be the dimensionality of the policy space and $k \ll n$ be the number of sampled trajectories used to approximate the derivatives. As shown in Section 6, the computation cost of full Hessian $O(n^2 + n^\omega)$ is quadratic in $n$ and strictly dominates all the others. In contrast, the costs of the identity matrix approximation $O(nk)$ and the Woodbury approximation $O(nk + k^\omega)$ are both linear in $n$. The Woodbury method offers an option to get a more accurate Hessian at low additional cost.

## 9  Conclusion

This paper considers learning a predictive model to address the missing parameters in sequential decision problems. We successfully extend decision-focused learning from optimization problems to MDP problems solved by deep reinforcement learning algorithms, where we apply sampling and low-rank approximation to Hessian matrix computation to address the associated computational challenges. All our results suggest that decision-focused learning can outperform two-stage approach by directly optimizing the final evaluation metric. The idea of considering sequential decision problems as differentiable layers also suggests a different way to solve online reinforcement learning problems, which we reserve as a future direction.

## Acknowledgement

This research was supported by MURI Grant Number W911NF-17-1-0370 and W911NF-18-1-0208. Chen and Perrault acknowledge support from the Center for Research on Computation and Society. Doshi-Velez acknowledges support from NSF CAREER IIS-1750358. The computations in this paper were run on the FASRC Cannon cluster supported by the FAS Division of Science Research Computing Group at Harvard University.

## Declaration of Interests

Doshi-Velez reports grants from the National Institutes of Health, and personal fees from Davita Kidney Care and Google Health via Adecco, outside of the submitted work. Tambe is jointly affiliated with Google Research India, outside of the submitted work. Wang, Shah, Chen, and Perrault declare no competing interests.

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
