# Appendix

## A    Proofs

**Theorem 1** (Policy gradient-based unbiased derivative estimate). *We follow the notation of Definition 1 and define* $\Phi_\theta(\tau, \pi) = \sum_{i=1}^h \sum_{j=i}^h \gamma^j R_\theta(s_j, a_j) \log \pi(a_i | s_i)$. *We have:*

$$\nabla_\pi J_\theta(\pi) = \mathbb{E}_{\tau \sim \pi, \theta} \left[ \nabla_\pi \Phi_\theta(\tau, \pi) \right] \implies \begin{aligned} \nabla_\pi^2 J_\theta(\pi) &= \mathbb{E}_{\tau \sim \pi, \theta} \left[ \nabla_\pi \Phi_\theta \cdot \nabla_\pi \log p_\theta^\top + \nabla_\pi^2 \Phi_\theta \right] \\ \nabla_{\theta\pi}^2 J_\theta(\pi) &= \mathbb{E}_{\tau \sim \pi, \theta} \left[ \nabla_\pi \Phi_\theta \cdot \nabla_\theta \log p_\theta^\top + \nabla_{\theta\pi}^2 \Phi_\theta \right] \end{aligned} \tag{9}$$

*First part of the proof (policy gradient theorem).* The first part of the proof follows the policy gradient theorem. We begin with definitions.

Let $\tau = \{s_1, a_1, s_2, a_2, \cdots, s_h, a_h\}$ be a trajectory sampled according to policy $\pi$ and MDP parameter $\theta$. Define $\tau_j = \{s_1, a_1, \cdots, s_j, a_j\}$ to be a partial trajectory up to time step $j$ for any $j \in [h]$. Define $G_\theta(\tau) = \sum_{j=1}^h \gamma^j R_\theta(s_j, a_j)$ to be the discounted value of trajectory $\tau$. Let $p_\theta(\tau, \pi)$ be the probability of seeing trajectory $\tau$ under parameter $\theta$ and policy $\pi$. Given MDP parameter $\theta$, we can compute the expected cumulative reward of policy $\pi$ by:

$$\begin{aligned} J_\theta(\pi) &= \mathbb{E}_{\tau \sim \pi, \theta} \, G_\theta(\tau) \\ &= \mathbb{E}_{\tau \sim \pi, \theta} \sum_{j=1}^h \gamma^j R_\theta(s_j, a_j) \\ &= \sum_{j=1}^h \mathbb{E}_{\tau \sim p_\theta(\tau, \pi)} \gamma^i R_\theta(s_j, a_j) \tag{13} \\ &= \sum_{j=1}^h \mathbb{E}_{\tau_j \sim p_\theta(\tau_j, \pi)} \gamma^j R_\theta(s_j, a_j) \tag{14} \\ &= \sum_{j=1}^h \int_{\tau_j} \gamma^j R_\theta(s_j, a_j) p_\theta(\tau_j, \pi) d\tau_j \end{aligned}$$

Equation 13 to Equation 14 uses the fact that we only need to sample up to time step $j$ in order to compute $\gamma^j R_\theta(s_j, a_j)$. Everything beyond time step $j$ does not affect the expectation up to time step $j$. We can compute the policy gradient by:

$$\begin{aligned} &\nabla_\pi J_\theta(\pi) \\ &= \nabla_\pi \sum_{j=1}^h \int_{\tau_j} \gamma^j R_\theta(s_j, a_j) p_\theta(\tau_j, \pi) d\tau_j \\ &= \sum_{j=1}^h \int_{\tau_j} \gamma^j R_\theta(s_j, a_j) \nabla_\pi p_\theta(\tau_j, \pi) d\tau_j \tag{15} \\ &= \sum_{j=1}^h \int_{\tau_j} \gamma^j R_\theta(s_j, a_j) p_\theta(\tau_j, \pi) \nabla_\pi \log p_\theta(\tau_j, \pi) d\tau_j \tag{16} \end{aligned}$$

where Equation 15 is because only the probability term is dependent on policy $\pi$, and Equation 16 is by $\nabla_\pi p_\theta = p_\theta \nabla_\pi \log p_\theta$.

We can now merge the integral back to an expectation over trajectory $\tau_j$ by merging the probability term $p_\theta$ and the integral:

$$
\begin{aligned}
\nabla_\pi J_\theta(\pi) &= \sum\nolimits_{j=1}^{h} \mathbf{E}_{\tau_j \sim p_\theta(\tau_j, \pi)} \left[ \gamma^j R_\theta(s_j, a_j) \nabla_\pi \log p_\theta(\tau_j, \pi) \right] \\
&= \sum\nolimits_{j=1}^{h} \mathbf{E}_{\tau \sim p_\theta(\tau, \pi)} \left[ \gamma^j R_\theta(s_j, a_j) \nabla_\pi \log p_\theta(\tau_j, \pi) \right] \\
&= \mathbf{E}_{\tau \sim p_\theta(\tau, \pi)} \left[ \sum\nolimits_{j=1}^{h} \gamma^j R_\theta(s_j, a_j) \nabla_\pi \log p_\theta(\tau_j, \pi) \right] \\
&= \mathbf{E}_{\tau \sim p_\theta(\tau, \pi)} \left[ \sum\nolimits_{j=1}^{h} \gamma^j R_\theta(s_j, a_j) \nabla_\pi \left( \sum\nolimits_{i=1}^{j} \log \pi(a_i \mid s_i) + \sum\nolimits_{i=1}^{j} \log p_\theta(s_i, a_i, s_{i+1}) \right) \right]
\end{aligned}
$$
(17)

$$
\begin{aligned}
&= \mathbf{E}_{\tau \sim p_\theta(\tau, \pi)} \left[ \sum\nolimits_{j=1}^{h} \gamma^j R_\theta(s_j, a_j) \sum\nolimits_{i=1}^{j} \nabla_\pi \log \pi(a_i \mid s_i) \right] \\
&= \mathbf{E}_{\tau \sim p_\theta(\tau, \pi)} \left[ \sum\nolimits_{j=1}^{h} \sum\nolimits_{i=1}^{j} \gamma^j R_\theta(s_j, a_j) \nabla_\pi \log \pi(a_i \mid s_i) \right] \\
&= \mathbf{E}_{\tau \sim p_\theta(\tau, \pi)} \left[ \sum\nolimits_{i=1}^{h} \sum\nolimits_{j=i}^{h} \gamma^j R_\theta(s_j, a_j) \nabla_\pi \log \pi(a_i \mid s_i) \right] \\
&= \mathbf{E}_{\tau \sim p_\theta(\tau, \pi)} \left[ \nabla_\pi \Phi_\theta(\tau, \pi) \right]
\end{aligned}
$$
(18)

where Equation 17 is by expanding the probability of seeing trajectory $\tau_j$ when parameter $\theta$ and policy $\pi$ are used, where the probability decomposes into the first term action probability $\pi(a_i \mid s_i)$, and the second term transition probability $p_\theta(s_i, a_i, s_{i+1})$, which is independent of policy $\pi$ and thus disappears. The last equation in Equation 18 connects back to the definition of $\Phi$ as defined in the statement of Theorem 1. $\Phi$ is easy to compute and easy to differentiate through. We can therefore sample a set of trajectories $\{\tau\}$ to compute the corresponding $\Phi$ and its derivative to get the unbiased policy gradient estimate. $\qquad\square$

*Second part of the proof (second-order derivatives).* Given the policy gradient theorem as we recall in the above derivation, we have:

$$
\nabla_\pi J_\theta(\pi) = \mathbf{E}_{\tau \sim p_\theta(\tau, \pi)} \left[ \nabla_\pi \Phi_\theta(\tau, \pi) \right]
$$
(19)

We can compute the derivative of Equation 19 by:

$$
\begin{aligned}
\nabla_\pi^2 J_\theta(\pi) &= \nabla_\pi \nabla_\pi J_\theta(\pi) \\
&= \nabla_\pi \mathbf{E}_{\tau \sim p_\theta(\tau, \pi)} \left[ \nabla_\pi \Phi_\theta(\tau, \pi) \right] \\
&= \nabla_\pi \int_\tau \nabla_\pi \Phi_\theta(\tau, \pi) p_\theta(\tau, \pi) d\tau \\
&= \int_\tau \left[ \nabla_\pi \Phi_\theta(\tau, \pi) \nabla_\pi p_\theta(\tau, \pi)^\top + \nabla_\pi^2 \Phi_\theta(\tau, \pi) p_\theta(\tau, \pi) \right] d\tau \\
&= \int_\tau \left[ \nabla_\pi \Phi_\theta(\tau, \pi) \nabla_\pi \log p_\theta(\tau, \pi)^\top + \nabla_\pi^2 \Phi_\theta(\tau, \pi) \right] p_\theta(\tau, \pi) d\tau \\
&= \mathbf{E}_{\tau \sim p_\theta(\tau, \pi)} \left[ \nabla_\pi \Phi_\theta(\tau, \pi) \nabla_\pi \log p_\theta(\tau, \pi)^\top + \nabla_\pi^2 \Phi_\theta(\tau, \pi) \right]
\end{aligned}
$$
(20)

(21)

where Equation 20 passes gradient inside the integral and applies chain rule. Equation 21 provides an unbiased estimate of the second-order derivative $\nabla_\pi^2 J_\theta(\pi)$.

Similarly, we can compute:

$$\begin{aligned}
\nabla^2_{\theta\pi} J_\theta(\pi) &= \nabla_\theta \nabla_\pi J_\theta(\pi) \\
&= \nabla_\theta \, \mathbf{E}_{\tau \sim p_\theta(\tau,\pi)} \left[ \nabla_\pi \Phi_\theta(\tau,\pi) \right] \\
&= \nabla_\theta \int_\tau \nabla_\pi \Phi_\theta(\tau,\pi) p_\theta(\tau,\pi) d\tau \\
&= \int_\tau \left[ \nabla_\pi \Phi_\theta(\tau,\pi) \nabla_\theta p_\theta(\tau,\pi)^\top + \nabla^2_{\theta\pi} \Phi_\theta(\tau,\pi) p_\theta(\tau,\pi) \right] d\tau \\
&= \int_\tau \left[ \nabla_\pi \Phi_\theta(\tau,\pi) \nabla_\theta \log p_\theta(\tau,\pi)^\top + \nabla^2_{\theta\pi} \Phi_\theta(\tau,\pi) \right] p_\theta(\tau,\pi) d\tau \\
&= \mathbf{E}_{\tau \sim p_\theta(\tau,\pi)} \left[ \nabla_\pi \Phi_\theta(\tau,\pi) \nabla_\theta \log p_\theta(\tau,\pi)^\top + \nabla^2_{\theta\pi} \Phi_\theta(\tau,\pi) \right] \quad (22)
\end{aligned}$$

Equation 21 and Equation 22 both serve as unbiased estimates of the corresponding second-order derivatives. We can sample a set of trajectories to compute both of them and get an unbiased estimate of the second-order derivatives. This concludes the proof of Theorem 1. $\qquad\square$

**Theorem 2** (Bellman-based unbiased derivative estimate). *We follow the notation in Definition 2 to define $J_\theta(\pi) = \frac{1}{2} \mathbf{E}_{\tau \sim \pi, \theta} \left[ \delta^2_\theta(\tau,\pi) \right]$. We have:*

$$\nabla_\pi J_\theta(\pi) = \mathbf{E}_{\tau \sim \pi, \theta} \left[ \delta \nabla_\pi \delta + \frac{1}{2} \delta^2 \nabla_\pi \log p_\theta \right] \implies \nabla^2_\pi J_\theta(\pi) = \mathbf{E}_{\tau \sim \pi, \theta} \left[ \nabla_\pi \delta \nabla_\pi \delta^\top + O(\delta) \right]$$

$$\nabla^2_{\theta\pi} J_\theta(\pi) = \mathbf{E}_{\tau \sim \pi, \theta} \left[ \nabla_\pi \delta \nabla_\theta \delta^\top + \left( \nabla_\pi \delta \nabla_\theta \log p_\theta^\top + \nabla_\pi \log p_\theta \nabla_\theta \delta^\top + \nabla^2_{\theta\pi} \delta \right) \delta + O(\delta^2) \right] \quad (10)$$

*First part of the proof (first-order derivative).* By the definition of $J_\theta(\pi) = \frac{1}{2} \mathbf{E}_{\tau \sim \pi, \theta} \left[ \delta^2(\tau,\pi) \right]$, we can compute its first-order derivative by:

$$\begin{aligned}
\nabla_\pi J_\theta(\pi) &= \nabla_\pi \frac{1}{2} \mathbf{E}_{\tau \sim \pi, \theta} \left[ \delta^2_\theta(\tau,\pi) \right] \\
&= \nabla_\pi \frac{1}{2} \int_\tau \delta^2_\theta(\tau,\pi) p_\theta(\tau,\pi) d\tau \\
&= \int_\tau \left[ p_\theta(\tau,\pi) \delta_\theta(\tau,\pi) \nabla_\pi \delta_\theta(\tau,\pi) + \frac{1}{2} \delta^2_\theta(\tau,\pi) \nabla_\pi p_\theta(\tau,\pi) \right] d\tau \\
&= \int_\tau \left[ \delta_\theta(\tau,\pi) \nabla_\pi \delta_\theta(\tau,\pi) + \frac{1}{2} \delta^2_\theta(\tau,\pi) \nabla_\pi \log p_\theta(\tau,\pi) \right] p_\theta(\tau,\pi) d\tau \\
&= \mathbf{E}_{\tau \sim \pi, \theta} \left[ \delta_\theta(\tau,\pi) \nabla_\pi \delta_\theta(\tau,\pi) + \frac{1}{2} \delta^2_\theta(\tau,\pi) \nabla_\pi \log p_\theta(\tau,\pi) \right] \quad (23)
\end{aligned}$$

$\qquad\square$

*Second part of the proof (second-order derivative).* Given Equation 23, we can further compute the second-order derivatives by:

$$\begin{aligned}
\nabla^2_\pi J_\theta(\pi) &= \nabla_\pi \nabla_\pi J_\theta(\pi) \\
&= \nabla_\pi \mathbf{E}_{\tau \sim \pi, \theta} \left[ \delta_\theta(\tau,\pi) \nabla_\pi \delta_\theta(\tau,\pi) + \frac{1}{2} \delta^2_\theta(\tau,\pi) \nabla_\pi \log p_\theta(\tau,\pi) \right] \\
&= \nabla_\pi \int_\tau \left[ \delta_\theta(\tau,\pi) \nabla_\pi \delta_\theta(\tau,\pi) + \frac{1}{2} \delta^2_\theta(\tau,\pi) \nabla_\pi \log p_\theta(\tau,\pi) \right] p_\theta(\tau,\pi) d\tau \\
&= \int_\tau \left( \nabla_\pi \delta \nabla_\pi \delta^\top + \delta \nabla^2_\pi \delta + \delta \nabla \log p_\theta \nabla_\pi \delta^\top + \frac{1}{2} \delta^2 \nabla^2 \log p_\theta \right) p_\theta \\
&\qquad + \left( \delta \nabla_\pi \delta(\tau,\pi) + \frac{1}{2} \delta^2 \nabla_\pi \log p_\theta \right) p_\theta \nabla \log p_\theta^\top \quad d\tau \\
&= \mathbf{E}_{\tau \sim \pi, \theta} \left[ \nabla_\pi \delta \nabla_\pi \delta^\top + \delta \nabla^2_\pi \delta + \delta \nabla \log p_\theta \nabla_\pi \delta^\top + \delta \nabla_\pi \delta(\tau,\pi) \nabla \log p_\theta^\top + O(\delta^2) \right] \\
&= \mathbf{E}_{\tau \sim \pi, \theta} \left[ \nabla_\pi \delta \nabla_\pi \delta^\top + O(\delta) \right]
\end{aligned}$$

Similarly, we have:

$$
\begin{aligned}
\nabla_{\theta\pi}^2 J_\theta(\pi) &= \nabla_\theta \nabla_\pi J_\theta(\pi) \\
&= \nabla_\theta\, \mathbf{E}_{\tau\sim\pi,\theta} \left[ \delta_\theta(\tau,\pi)\nabla_\pi \delta_\theta(\tau,\pi) + \frac{1}{2}\delta_\theta^2(\tau,\pi)\nabla_\pi \log p_\theta(\tau,\pi) \right] \\
&= \nabla_\theta \int_\tau \left[ \delta_\theta \nabla_\pi \delta_\theta + \frac{1}{2}\delta_\theta^2 \nabla_\pi \log p_\theta \right] p_\theta d\tau \\
&= \int_\tau \left( \nabla_\pi \delta \nabla_\theta \delta^\top + \delta \nabla_{\theta\pi}^2 \delta + \delta \nabla_\pi \log p_\theta \nabla_\theta \delta^\top + \frac{1}{2}\delta^2 \nabla_{\theta\pi}^2 \log p_\theta \right) p_\theta \\
&\qquad + \left( \delta \nabla_\pi \delta + \frac{1}{2}\delta^2 \nabla_\pi \log p_\theta \right) p_\theta \nabla_\theta \log p_\theta^\top \quad d\tau \\
&= \mathbf{E}_{\tau\sim\pi,\theta} \left[ \nabla_\pi \delta \nabla_\theta \delta^\top + \delta \nabla_{\theta\pi}^2 \delta + \delta \nabla_\pi \log p_\theta \nabla_\theta \delta^\top + \delta \nabla_\pi \delta \nabla_\theta \log p_\theta^\top + O(\delta^2) \right] \\
&= \mathbf{E}_{\tau\sim\pi,\theta} \left[ \nabla_\pi \delta \nabla_\theta \delta^\top + \left( \nabla_{\theta\pi}^2 \delta + \nabla_\pi \log p_\theta \nabla_\theta \delta^\top + \nabla_\pi \delta \nabla_\theta \log p_\theta^\top \right) \delta + O(\delta^2) \right]
\end{aligned}
\tag{24}
$$

which concludes the proof. $\qquad\qquad\square$

## B  Additional Discussions of Decision-focused Learning

In this section, we provide additional discussions of applying decision-focused learning to MDPs problems.

### B.1  Smoothness of the Optimal Policy Derived From Reinforcement Learning Solver

In Equation 8, we compute the gradient of the final evaluation metric with respect to the predictive model by applying chain rule. This implicitly requires each individual component in the chain rule to be well-defined. Specifically, the mapping from the MDP parameters to the optimal policy needs to be smooth so that we can compute a meaningful derivative of the policy with respect to the MDP parameters. However, this smoothness requirement is only required in the training time to make the gradient computation available. Once the training is finished, there is no restriction on the policy and the corresponding solver. This smoothness requirement does not restrict the kind of problems that we can solve. We just need to find a solver that can give a smooth policy to ensure the differentiability at training time, e.g., soft actor critic and soft Q learning.

Specifically, the assumption on smooth policy is similar to the idea of soft Q-learning [12] and soft actor-critic [13] proposed by Haarnoja et al. Soft Q-learning relaxes the Bellman equation to a soft Bellman equation to make the policy smoother, while soft actor-critic adds an entropy term as regularization to make the optimal policy smoother. These relaxed policy not only can make the training smoother as stated in the above papers, but also can allow back-propagation through the optimal policy to the input MDP parameters in our paper. These benefits are all due to the smoothness of the optimal policy. Similar issues arise in decision-focused learning in discrete optimization, with Wilder et al. [36] proposing to relaxing the optimal solution by adding a regularization term, which serves as the same purpose as we relax our optimal policy in the sequential decision problem setting.

### B.2  Unbiased Second-order Derivative Estimates

As we discuss in Section 6, correctly approximating the second-order derivatives is the crux of our algorithm. Incorrect approximation may lead to incorrect gradient direction, which can further lead to divergence. Since the second-order derivative formulation as stated in Theorem 1 and Theorem 2 are both unbiased derivative estimate. However, their accuracy depends on how many samples we use to approximate the derivatives. In our experiments, we use 100 sampled trajectories to approximate the second-order derivatives across three domains. The number of samples required to get a sufficiently accurate derivative estimate may depend on the problem size. Larger problems may require more samples to get a good derivative estimate, but more samples also implies more computation cost required to run the back-propagation.

In practice, we find that normalization effect of the Hessian term as discussed in Section 6 is very important to reduce the variance caused by the incorrect derivative estimate. Additionally, we also

notice that adding a small additional predictive loss term to run back-propagation can stabilize the training process because the predictive loss does not suffer from sampling variance. This is why we add a weighted predictive loss to the back-propagation in Algorithm 1.

## B.3 Impact of Optimality in the Forward Pass

In order to differentiate through the KKT conditions, we need the policy $\pi^*$ return by the reinforcement learning solver to be optimal in Figure 1. However, sub-optimal solution is often reached by the reinforcement learning solver and the optimality can impact the gradient computed from differentiating through the KKT conditions.

In this section, we analyze the impact of a sub-optimal policy produced by the reinforcement learning solver. When the problem is smooth, or more precisely when the function $J_\theta(\pi)$ is smooth around the optimal policy $\pi^*$, we can bound the gradients $\nabla^2_\pi J_\theta(\pi')$ and $\nabla_\pi J_\theta(\pi')$ computed in Equation 8 using a sub-optimal policy $\pi'$ by the gradients computed using the optimal policy $\pi^*$. Specifically, if the Hessian $\nabla^2_\pi J_\theta(\pi^*)$ is sufficiently far from singular, the difference between two gradients computed from sub-optimal and optimal policy using Equation 8 can be written as:

$$\left| \frac{d\,\mathrm{Eval}(\pi')}{d\pi}(\nabla^2_\pi J_\theta(\pi'))^{-1}\nabla^2_{\theta\pi}J_\theta(\pi')\frac{d\theta}{dw} - \frac{d\,\mathrm{Eval}(\pi^*)}{d\pi}(\nabla^2_\pi J_\theta(\pi^*))^{-1}\nabla^2_{\theta\pi}J_\theta(\pi^*)\frac{d\theta}{dw} \right|$$

which can be further bounded by applying telescoping sum to decompose the difference into linear combination of the difference in each individual gradient term. This suggests that when the smoothness condition of the above derivatives is met, we can bound the error incurred by sub-optimal policy.

## C Experimental Setup

In this section, we describe how we randomly generate the MDP problems and the corresponding missing parameters.

**Feature generation** Across all three domains, once the missing parameters are generated, we feed each MDP parameter into a randomly initialized neural network with two intermediate layers each with 64 neurons, and an output dimension size 16 to generate a feature vector of size 16 for the corresponding MDP parameter. For example, in the gridworld example, each grid cell comes with a missing reward. So the feature corresponding to this grid cell and the missing reward is generated by feeding the missing reward into a randomly initialized neural network to generate a feature vector of size 16 for this particular grid cell. We repeat the same process for all the parameters in the MDP problem, e.g., all the grid cells in the gridworld problem. The randomly initiated neural network uses ReLU layers as nonlinearity followed by a linear layer in the end. The generated features are normalized to mean 0 and variance 1, and we add Gaussian noise $\mathcal{N}(0, 1)$ to the features, with a signal noise ratio is $1 : 3$, to model that the original missing parameters may not be perfectly recovered from the noisy features. The predictive model we use to map from generated noisy features to the missing parameters is a single layer neural network with 16 neurons.

**Training parameters** Across all three examples, we consider the discounted setting where the discount factor is $\gamma = 0.95$. The learning rate is set to be $\alpha = 0.01$. The number of demonstrated trajectories is set to be 100 in both the random and near-optimal settings.

**Reinforcement learning solvers** In order to train the optimal policy, in the gridworld example, we use tabular value-iteration algorithm to learn the Q value of each state action pair. In the snare finding and the TB problems, since the state space is continuous, we apply DDQN [27, 32] to train the Q function and the corresponding policy, where we use a neural network with two intermediate layers each with 64 neurons to represent the function approximators of the Q values. There is one exception in the runtime plot of the snare finding problem in Figure 3(c), where the full Hessian computation is infeasible when a two layer neural network is used. Thus we use an one layer neural network with 64 neurons only to test the runtime of different Hessian approximations.

## C.1 Gridworld Example With Missing Rewards

**Problem setup**  We consider a $5 \times 5$ Gridworld environment with unknown rewards as our MDP problems with unknown parameters. The bottom left corner is the starting point and the top right corner is a safe state with a high reward drawn from a normal distribution $\mathcal{N}(5, 1)$. The agent can walk between grid cells by going north, south, east, west, or deciding to stay in the current grid cells. So the number of available actions is 5, while the number of available states is $5 \times 5 = 25$.

The agent collects reward when the agent steps on each individual grid cell. There is $20\%$ chance that each intermediate grid cell is a cliff that gives a high penalty drawn from another normal distribution $\mathcal{N}(-10, 1)$. All the other $80\%$ of grid cells give rewards drawn from $\mathcal{N}(0, 1)$. The goal of the agent is to collect as much reward as possible. We consider a fixed time horizon case with 20 steps, which is sufficient for the agent to go from bottom left to the top right corner.

**Training details**  Within each individual training step for each MDP problem with missing parameters, we first predict the rewards using the predictive model, and then solve the resulting problem using tabular value-iteration. We run in total 10000 iterations to learn the Q values, which are later used to construct the optimal policy. To relax the optimal policy given by the RL solver, we relax the Bellman equation used to run value-iteration by relaxing all the argmax and max operators in the Bellman equation to softmax with temperature 0.1, i.e., we use SOFTMAX$(0.1 \cdot \text{Q-values})$ to replace all the argmax over Q values. The choice of the tempreratue 0.1 is to make sure that the optimal policy is smooth enough but the relaxation does not impact the optimal policy too much as well.

**Random and near-optimal trajectories generation**  To generate the random trajectories, we have the agent randomly select actions between all actions. To generate the near-optimal trajectories, we replace the softmax with temperature 0.1 by softmax with temperature 1 and train an RL agent using ground truth reward values by 50000 value-iterations to get a near-optimal policy. We then use the trained near-optimal policy to generate 100 independent trajectories as our near-optimal demonstrated trajectories.

## C.2 Snare Finding Problem With Missing Transition Probability

**Problem setup**  In the snare finding problem, we consider a set of 20 sites that are vulnerable to poaching activity. We randomly select $20\%$ of the sites as high-risk locations where the probability of having a poacher coming and placing a snare is randomly drawn from a normal distribution $\mathcal{N}(0.8, 0.1)$, while the remaining $80\%$ of low-risk sites with probability $\mathcal{N}(0.1, 0.05)$ having a poacher coming to place a snare. These transition probabilities are not known to the ranger, and the ranger has to rely on features of each individual site to predict the corresponding missing transition probability.

We assume the maximum number of snare is 1 per location, meaning that if there is a snare and it has not been removed by the ranger, then the poacher will not place an additional snare there until the snare is removed. The ranger only observes a snare when it is removed. Thus the MDP problem with given parameters is partially observable, where the state maintained by the ranger is the belief of whether a site contains a snare or not, which is a fractional value between 0 and 1 for each site.

The available actions for the ranger are to select a site from 20 sites to visit. If there is a snare in the location, the ranger successfully removes the snare and gets reward 1 with probability 0.9, and otherwise the snare remains there with a reward $-1$. If there is no snare in the visited site, the ranger gets reward $-1$. Thus the number of actions to the ranger is 20, while the state space is continuous since the ranger uses continuous belief as the state.

**Training details**  To solve the optimal policy from the predicted parameters, we run DDQN with 1000 iterations to collect random experience and 10000 iterations to train the model. We use a replay buffer to store all the past experience that the agent executed before. To soften the optimal policy, we also use a relaxed Bellman equation as stated in Section C.2. Because the cumulative reward and the corresponding Q values in this domain is relatively smaller than the Gridworld domain, we replace all the argmax and max operators by softmax with a larger temperature 1 to reflect the relatively smaller reward values.

**Random and near-optimal trajectories generation**    Similar to Section C.1, we generate the random trajectories by having the agent choose action from all available actions uniformly at random. To generate the near-optimal trajectories, we replace all the softmax with temperature 1 by softmax with temperature 5, and we use the ground truth transition probabilities to train the RL agent by DDQN using 50000 iterations to generate a near-optimal policy. The near-optimal trajectories are then generated by running the trained near-optimal policy for 100 independent runs.

### C.3    Tuberculosis With Missing Transition Probability

**Problem setup**    There are a total of 5 patients who need to take their medication at each time-step for 30 time-steps. For each patient, there are 2 states – non-adhering (0), and adhering (1). The patients are assumed to start from a non-adhering state. Then, in subsequent time-steps, the patients' states evolve based on their current state and whether they were intervened on by a healthcare worker according to a transition matrix.

The raw transition probabilities for different patients are taken from [18].[5] However, these raw probabilities do not record a patient's responsiveness to an intervention. To incorporate the effect of intervening, we sample numbers from $U(0, 0.4)$, and (a) add them to the probability of adhering when intervened on, and (b) subtract them from the probability of adhering when not. Finally, we clip the probabilities to the range of $[0.05, 0.95]$ and re-normalize. We use the raw transition probabilities and the randomly generated intervention effect to model the behavior of our synthetic patients and generate all the training trajectories accordingly. The entire transition matrix for each patient is then fed as an input to the feature generation network to generate features for that patient. In this example, we assume the transition matrices to be missing parameters, and try to learn a predictive model to recover the transition matrices from the generated features using either two-stage or various decision-focused learning methods as discussed in the main paper.

Given the synthetic patients, we consider a healthcare worker who has to choose one patient at every time-step to intervene on. However, the healthcare worker can only observe the 'true state' of a patient when she intervenes on them. At every other time, she has a 'belief' of the patient's state that is constructed from the most recent observation and the predicted transition probabilities. Therefore, the healthcare worker has to learn a policy that maps from these belief states to the action of whom to intervene on, such that the sum of adherences of all patients is maximised over time. The healthcare worker gets a reward of 1 for an adhering patient and 0 for a non-adhering one. Like Problem C.2, this problem has a continuous state space (because of the belief states) and discrete action space.

**Training details**    Same as Section C.2.

**Random and near-optimal trajectories generation**    Similar to Section C.2, we generate the random trajectories by having the agent choose action from all available actions uniformly at random. To generate the near-optimal trajectories, we replace all the softmax with temperature 5 by softmax with temperature 20,[6] and we use the ground truth transition probabilities to train the RL agent by DDQN using $100,000$ iterations to generate a near-optimal policy. The near-optimal trajectories are then generated by running the trained near-optimal policy for 100 independent runs.

## D    Additional Experiment Results

**Tuberculosis Adherence**    The results for this problem can be found in Table 1, and the training curves can be found in Figure 4. While the standard errors associated with the results *seem* very large, this is in large part because of the way in which we report them. To keep it consistent with other problems, we average the absolute OPE scores for each method across multiple problem instances. However, in the TB case, each problem instance can be very different because the patients in each

---

[5]The raw transition probabilities taken from [18] are only used to generate synthetic patients.

[6]The reason that we use a relatively larger temperature is because the range of the cumulative reward in TB domain is smaller than the previous two domains. In TB domain, the patients could change from non-adhering back to adhering even if there is no intervention, while in contrast, a snare placed in a certain location will not be removed until the ranger visits the place. In other words, the improvement that intervention can introduce is relatively limited compared to the snare finding domain. Thus even though the cumulative reward in TB domain is larger than the previous two domains, the range is smaller and thus we need a larger temperature.

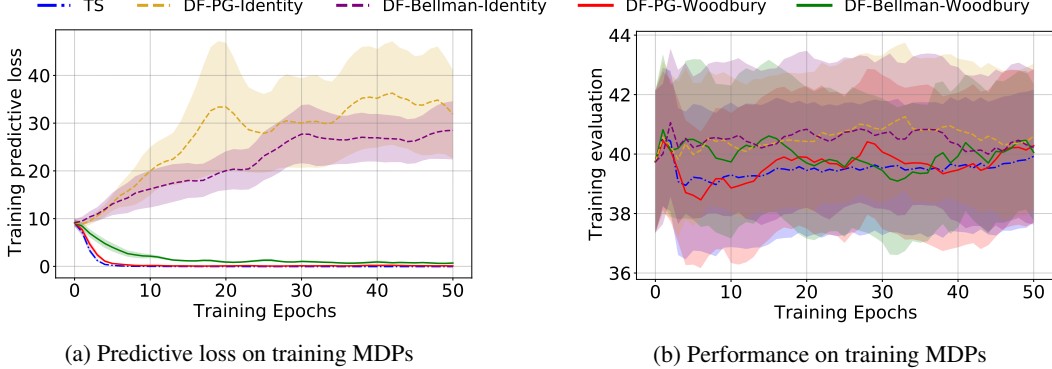

(a) Predictive loss on training MDPs

(b) Performance on training MDPs

Figure 4: Learning curves of for the TB problem with random trajectories.

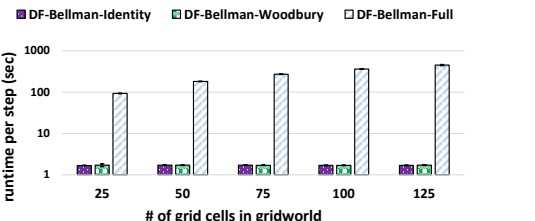

(a) Backpropagation runtime per gradient step of Bellman equation-based decision-focused learning using different Hessian approximations in the gridworld problem.

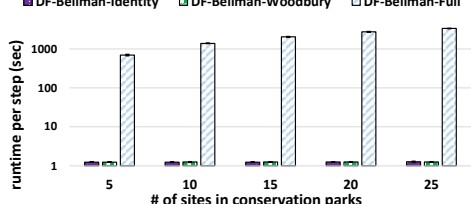

(b) Backpropagation runtime per gradient step of Bellman equation-based decision-focused learning using different Hessian approximations in the snare finding problem.

Figure 5: We compare the backpropagation runtime of decision-focused learning methods using bellman optimality with different Hessian approximations. We can see that the runtime of both identity and Woodbury methods largely outperform the runtime of full Hessian computation, demonstrating the importance of the Hessian approximation. Additionally, the runtime of Woodbury method using low-rank approximation is similar to the runtime of identity method. Woodbury method provides a more accurate approximation with a similar runtime.

of these instances are sampled from the transition probabilities previously studied in [18] that have diverse patient behaviour. As a result, the baseline OPE values vary widely across different problem instances, causing a larger variation in Figure 4(b) and contributing as the major source of standard deviation.

**Computation cost of Bellman equation-based decision-focused methods**    We additionally compare the runtime of the operation of backpropagation per gradient step of Bellman equation-based decision-focused learning using different Hessian approximations. This is the runtime required to compute the gradient in the backward pass. We can see that the runtime of methods using identity and Woodbury methods are much smaller than the runtime of full Hessian approximation. This matches to our analysis in Section 6 and the experimental results in Figure 2(c) and Figure 3(c).

**Choice of regularizer $\lambda$ in Algorithm 1 and ablation study**    We ran an ablation study by varying the regularization constant $\lambda$ in Algorithm 1 using the snare-finding problem. The experimental result is shown in Figure 6. The role of regularization in Algorithm 1 is to help resolve the non-convexity issue of the off-policy evaluation (OPE) objective. Decision-focused learning methods can easily get trapped by various local minima due to the non-convexity of the OPE metric. Adding a small two-stage loss can improve the convexity of the optimizing objective and thus help improve the performance as well. We can see that adding small amount of regularization can usually help improve the overall performance in both cases with random and near-optimal trajectories. However, adding too much regularization in Algorithm 1 can push decision-focused learning toward two-stage approach,

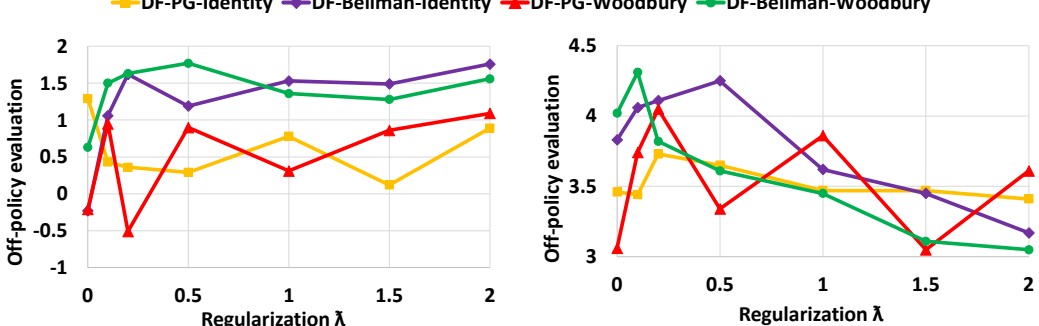

(a) The off-policy evaluation performance of different regularization $\lambda$ in snare-finding problem with random trajectories.

(b) The off-policy evaluation performance of different regularization $\lambda$ in snare-finding problem with near-optimal trajectories.

Figure 6: Ablation study of different regularization $\lambda$ in Algorithm 1 on the snare-finding problem using different decision-focused learning methods.

which can degrade the performance sometimes. The right amount of regularization is critical to balance between the issue of convexity and the optimizing objective.

# E   Computing Infrastructure

All experiments except were run on a computing cluster, where each node is configured with 2 Intel Xeon Cascade Lake CPUs, 184 GB of RAM, and 70 GB of local scratch space. Within each experiment, we did not implement parallelization nor use GPU, so each experiment was purely run on a single CPU core.