# OpenReview forum: "Learning MDPs from Features: Predict-Then-Optimize for Sequential Decision Making by Reinforcement Learning"
_NeurIPS.cc/2021/Conference — NeurIPS 2021 Spotlight_

### Official Review · Reviewer_gBYr · 2021-07-11

**Rating:** 7
**Confidence:** 3

**Summary:**

In this paper, the authors consider the problem of learning a predictive model for MDPs with missing parameters. Instead of a standard two-stage approach, an end-to-end approach based on differentiating through an optimality condition is proposed. To address the associated computation issues, a sampling method and a low-rank approximation method are introduced. Empirical evaluations on three synthetic settings are provided to show that the end-to-end approach outperforms the two-stage approach.

**Limitations And Societal Impact:**

The authors raised some interesting hypotheses related to the design choices in section 8. It would be great if they can also provide concrete analyses on those hypotheses.

**Main Review:**

Post-rebuttal comment: I thank the authors for answering my questions and providing additional ablations. I will keep my score.

Originality: The problem setting is not entirely original but is interesting as it has connections to different areas such as meta-RL. I enjoyed reading the end-to-end approach which tackles a core problem of differentiating a policy’s offline off-policy evaluation with respect to its parameters. Differentiating through an optimality condition is a popular method in related works on embedding an optimization layer in a neural network model, for example [1]. The extension to MDPs is definitely interesting and I think the authors did a good job addressing the technical challenges from such an extension.

Quality: The method is technically sound. I have two questions about the details in algorithm 1.
1. The forward step (line 4) requires obtaining an optimal policy on the predicted MDP problem. Since the KKT condition also applies to the optimal policy, what is the impact if running a model-free RL algorithm only produces a sub-optimal policy?

2. The gradient step (line 6) mentioned using the predictive loss as a regularizer. Did you do an ablation study on how important this regularization is? What are some high-level explanations on why this regularization is necessary?

Clarity: This paper is well-written and easy to follow. Good job!

Significance: I think the idea presented in this work is of interest to the broad NeurIPS community. One weakness is that the three empirical settings are synthetic which may hide some potential issues of the algorithm. For example, related to the first question in the quality section, the robustness of the algorithm should be tested in more complex settings where obtaining an optimal policy is difficult.

[1]: End to end learning and optimization on graphs, Bryan Wilder, Eric Ewing, Bistra Dilkina, Milind Tambe, NeurIPS 2019


**Time Spent Reviewing:**

2

---

> ### Author Response · Authors · 2021-08-10
> **Error bound analysis on the gradient computed from sub-optimal policies + an additional ablation study on the regularization**
>
> We thank the reviewer for all the constructive feedback. Please find our response to the questions related to Algorithm 1 below.
>
> **Question**: The forward step (line 4) requires obtaining an optimal policy on the predicted MDP problem. Since the KKT condition also applies to the optimal policy, what is the impact if running a model-free RL algorithm only produces a sub-optimal policy?
>
> **Answer**: Given a sub-optimal policy produced by a model-free solver, when the problem is smooth, or more precisely when the function $J_\theta(\pi)$ is smooth around the optimal policy $\pi^*$, it is possible to bound the gradients $\nabla^2_\pi J_\theta(\pi')$ and $\nabla_\pi J_\theta(\pi')$ computed in Equation (8) using a sub-optimal policy $\pi'$ and the gradients computed using the optimal policy $\pi^*$. Specifically, if the Hessian $\nabla^2_\pi J_\theta(\pi^*)$ is sufficiently far from singular, we can bound the difference between two gradients by applying telescoping sum to estimate the difference:
> $$\left| \frac{d ~ \text{Eval}(\pi')}{d \pi} (\nabla^2_{\pi} J_{\theta}(\pi'))^{-1} \nabla^2_{\theta \pi} J_{\theta}(\pi') \frac{d \theta}{d w}
>     - \frac{d ~ \text{Eval}(\pi^*)}{d \pi} (\nabla^2_{\pi} J_{\theta}(\pi^*))^{-1} \nabla^2_{\theta \pi} J_{\theta}(\pi^*) \frac{d \theta}{d w} \right|$$
>
> This suggests that when the above smoothness conditions are met, we can bound the error incurred by sub-optimal policy. We will add this analysis to the appendix.
>
> **Question**: The gradient step (line 6) mentioned using the predictive loss as a regularizer. Did you do an ablation study on how important this regularization is? What are some high-level explanations on why this regularization is necessary?
>
> **Answer**: The reason that predictive loss is used as a regularizer is that predictive loss is more convex than the final off-policy evaluation metric. The use of predictive loss can help reduce the non-convexity of the optimizing objective and thus the learning process. Throughout the experiments, we used a fixed amount of regularization $\lambda=0.1$.
>
> We ran an additional ablation study by varying the regularization constant $\lambda$ in Algorithm 1. The experimental result can be found in:
>
> https://www.dropbox.com/s/6p4oh92uilj9tpa/ablation.pdf?dl=0.
>
> Specifically, we can see that adding small amount of regularization can usually help improve the overall performance. We thank the reviewer for the constructive suggestion and we will add the ablation study to the appendix.

---

### Official Review · Reviewer_R1HM · 2021-07-15

**Rating:** 7
**Confidence:** 3

**Summary:**

The paper proposes a learning problem in which a partially defined MDP is provided with missing parameter values, particularly a subset of the transition probabilities and reward values. Input features defining each MDP are provided, along with sample trajectories from its stochastic policy. The goal is to learn to predict the missing parameter values, by targeting the sample trajectories given the MDP features. The prediction model is trained end-to-end across several stages: Prediction of missing MDP parameters from feature values, followed by reinforcement learning to solve the MDP for an optimal policy, and finally evaluation of the policy as an expectation of some loss over its trajectories. The primary challenge is differentiation through the reinforcement learning layer, which is achieved by differentiating through the optimality conditions of the decision problem. Computational intractability of differentiating with respect to large state and action spaces is handled with one of multiple proposed gradient approximations.


**Limitations And Societal Impact:**

Limitations of the methods are discussed in the paper.

**Main Review:**

[Post-rebuttal comments]
Thank you for your response. I have no further questions.

[Significance]
The paper offers clearly significant contributions, particularly with respect to the approach to differentiating the MDP layer. However, the significance of the application (learning missing parameter values) is not entirely clear to this reviewer. Overall, the paper does not provide much motivation for the significance of this problem and the need for advanced solution methods. The paper could benefit from a discussion along these lines.

[Novelty]
The paper uses argmin differentiation techniques applied to a novel setting and proposes clever approximations to handle the very large resulting computational problem.

[Readability & Technical Quality]
The paper is very well written, easy to follow, makes economic use of symbolism, and introduces concepts in a reader-friendly order. It is easy to understand despite several interacting concepts in a complex model. The experimental results and caveats are described very nicely. However, the description of the case studies in the experimental portion could be more detailed. It is somewhat unclear the exact specifications of the reinforcement learning tasks; even a small image, if available, would be helpful.

A few specific comments on aspects that are unclear:

 - Are there only a portion of missing parameters in this setting, or are all parameters missing? If only a portion is assumed to be missing, shouldn't each case study come with a description of which parameters are missing, and in what proportion? E.g., 70,80% or reward values missing. If this is the case, then the significance of the problem being solved is in more need of explanation and motivation.

 - The specific form of the underlying models that produce trajectories in the case studies -- are these specified? I.e., the architecture of the policy network.

 - Perhaps something should be said about the initialization of the learned parameters in the case studies

 - It seems that it is not mentioned until the appendix, that the underlying deep Q learning models in the case studies require their policy functions to be replaced with a smoothed approximation in order for the methodology to be applied. The appearance of Q-learning models in the main text results is confusing since the methods described prior assume a continuous policy. If not already, this should be clarified in the main text and the smoothed relaxations should be specified with some mathematical detail, possibly in the appendix.

 - For clarity, the nature of the loss functions l targeting the trajectories in Eqn (3) should be described.

The methods introduced are very well motivated on a technical level; the difficulties are thoroughly explained and novel approaches are used to handle them. Thorough proofs are provided, and the technical approach is largely convincing.

 [Minor Comments]

 - In Eqn (5), if appropriate - the equation seems like an abuse of notation, slightly confusing. Perhaps putting an argmax on the RHS or splitting the expression into 2 lines would be more clear.

**Time Spent Reviewing:**

~4

---

> ### Author Response · Authors · 2021-08-10
> **Explanation of missing parameters**
>
> We thank the reviewer for all the constructive feedback and suggestions for improvement. We will make sure to clarify any confusions pointed out by the reviewer to improve the paper.
>
> **Question**: Are there only a portion of missing parameters in this setting, or are all parameters missing?
>
> **Answer**: We do not put any restriction on the MDP parameter $\theta$, where the MDP parameter can affect multiple things in the problem, e.g., reward function and transition probabilities. Our approach can also handle an arbitrary portion of missing parameters by letting our parameter $\theta$ be the unknown portion of the MDP problem.
> For example, in our grid world problem, all reward parameters are missing but the transition probabilities are known. In the snare finding problem and the Tuberculosis problem, all transition probabilities are missing but the immediate reward is known.
>
> **Question**: The specific form of the underlying models that produce trajectories in the case studies -- are these specified? I.e., the architecture of the policy network.
>
> **Answer**: In snare finding and Tuberculosis problems that use DQN, we use a fully connected neural network with two layers, each with 64 neurons, to generate the near-optimal policy.
>
> **Comment**: Explanation of initialization, smoothed relaxations, and the definition of the loss functions $l$ targeting the trajectories in Eqn (3).
>
> **Answer**: The learned parameters are randomly initialized by a neural network. We will add more explanation to the paper. We will also elaborate more on the smoothed relaxation and the loss function in Eqn (3) in the main paper and add mathematical detail to the appendix.

---

> > ### Comment · Reviewer_R1HM · 2021-08-25
> > **Thank you for your response**
> >
> > I wanted to thank you for your response. They were very much appreciated.

---

### Official Review · Reviewer_HiZT · 2021-07-18

**Rating:** 7
**Confidence:** 3

**Summary:**

The authors use the predict-then-optimize framework to tackle prediction of MDP parameters based on input features. They achieve this by formulating MDP optimality KKT conditions and differentiating through them. In addition, the paper provides an ubiased sampling method for estimation of the KKT derivatives and a low-rank approximation to the high-dimensional sample-based derivatives. The authors name their approach decision-focused learning.

**Ethical Concerns:**

There are no ethical concerns.

**Limitations And Societal Impact:**

yes

**Main Review:**

# Summary

The authors use the predict-then-optimize framework to tackle prediction of MDP parameters based on input features. They achieve this by formulating MDP optimality KKT conditions and differentiating through them. In addition, the paper provides an ubiased sampling method for estimation of the KKT derivatives and a low-rank approximation to the high-dimensional sample-based derivatives. The authors name their approach decision-focused learning.

# Detailed Summary

## The Good

I think that the proposed method is quite novel. The paper addresses the computational issues of inferring MDP parameters when differentiating through KKT conditions. The experiment section showcases the benefits of applying the method proposed in the paper. Each component of the final algorithm is relatively well-argued (except the metric chosen).

## The Bad

I think that the assumptions on the MDP parameters $\theta$ and their restrictions have not been well discussed, the actual motivation for the problem statement is a bit lacking, since this is a relatively untypical problem setting that doesn't appear in RL that much, I think that the authors should motivate it better (with where this is applicable and what is the benefit as opposed to vanilla RL).

Furthermore, I think that the experiment section is weak, it mostly showcases the method in toy tasks, but as the authors note, and I understand,  learning MDPs from data is hard, so I wouldn't dwell on this that much.

## Detailed Comments


l.105: would be helpful here to give an example of what kind of features x are to be expected. Are they images, observations, sequences of observations etc. This description should argue why is this method useful.


l.109: do you mean rather that optimal trajectories are hidden, the trajectories from $\pi^*$?
l.109: please state explicitly that the predictive model is parametrized by parameters $\omega$, eg neural network.


l.115: I think that the choice of evaluation metric warrants a discussion of why this is a good metric, because people (for example myself) are perhaps not familiar with the work of Futoma et al.

l.141: by intuition, I would say that the assumption that the policy is smooth in $\theta$ is very restrictive in the kind of problems that you can solve, eg. just turning transitions on or off by setting transition probabilities  to 0. Maybe you can comment a bit more on the restrictions imposed by this assumption.

Def. 1: I have a slight issue with notation here, oftentimes in RL papers $J$ is used to denote the policy return as far as I know, so perhaps another notation here would be more suitable to ddenote the Bellman error under policy.

l.161: you are saying that the reward depends on parameter $\theta$, this is already restricting the kind of problems that you can solve, since you are talking about very specific MDP parameters. Otherwise, if the parameters affect the transition probabilities only, or the discount factor, the one-step reward might not be affected, but the solution of the MDP is.

Sec. 7: the set of experiments is rather small, and they are mostly toy experiments.

l.258:  I am not familiar with this snare-finding  MDP setting, how is this not a bandit problem where each arm is the site to be visited and the reward is sampled with some probability? Here the reward doesn't depend on the set of features?

l.277:  in this example, the reward doesn't depend on the set of features? Please state explicitly what is the set of features here.

Fig. 2: the color for the DF-PG-Identity setting needs to be changed, currently it's barely visible in the performance plots.

## Related Work

Additional related work that concerns itself with inferring parameters of MDP in constrained setting that you should consider adding to related work:

http://proceedings.mlr.press/v139/vlastelica21a.html



## Conclusion

All in all, I think it's a good paper that could use a few tweaks.






**Time Spent Reviewing:**

2

---

> ### Author Response · Authors · 2021-08-10
> **Response to detailed comments**
>
> We thank the reviewer for the constructive feedback and suggestions. We will take into account the comments and our responses correspondingly when polishing the paper. Please find our response to the questions and comments below
>
> **Comment**: [l.105] would be helpful here to give an example of what kind of features x are to be expected. Are they images, observations, sequences of observations etc. This description should argue why is this method useful.
>
> **Answer**: In the Tuberculosis problem, the features can be some patients' features that are correlated to how adherent the patients are, i.e., the unknown transition probability in the MDP. In the snare finding problem, the features can be the environmental features and some description of the conservation site, e.g., terrain features and GIS information, which are also correlated to how the snares are placed and the corresponding transition probability in the MDP. We thank the reviewer for the feedback, and we will update the paper to provide some examples of features.
>
> **Question**: [l.109] do you mean rather that optimal trajectories are hidden, the trajectories from $\pi^*$?
>
> **Answer**: Yes, trajectories are hidden in the test MDPs. The trajectories can come from any behavioral policy that is not necessarily the optimal one. We will make this clear in our final draft.
>
> **Comment**: [l.115] The choice of evaluation metric warrants a discussion of why this is a good metric, because people are perhaps not familiar with the work of Futoma et al.
>
> **Answer**: We thank the reviewer for the feedback. We will add more details to elaborate the evaluation metric used in the paper.
>
> **Question**: [l.141] by intuition, I would say that the assumption that the policy is smooth in $\theta$ is very restrictive in the kind of problems that you can solve, eg. just turning transitions on or off by setting transition probabilities to $\theta$. Maybe you can comment a bit more on the restrictions imposed by this assumption.
>
> **Answer**: The requirement that the policy needs to be smooth in $\theta$ is only required at training time, which is to ensure the end-to-end differentiability at training time. At testing time, there is no restriction on the policy. Therefore, we can relax the strict policy at training time to get the differentiability and run end-to-end gradient descent. At testing time, we can use the strict policy without relaxation or use a policy with tunable smoothness. Such smoothness requirement does not restrict the kind of problems that we can solve. We just need to find a solver that can give a smooth policy to ensure the differentiability at training time, e.g., soft actor critic and soft Q learning.
>
> **Comment**: [Def. 1] I have a slight issue with notation here, oftentimes in RL papers is used to denote the policy return as far as I know, so perhaps another notation here would be more suitable to denote the Bellman error under policy.
>
> **Answer**: We chose to use $J$ to align with the notations in the policy gradient method because the later analysis applies to both methods. We will check with the literature to see if there is a better choice of notation.
>
> **Question**: [l.161] you are saying that the reward depends on parameter, this is already restricting the kind of problems that you can solve, since you are talking about very specific MDP parameters. Otherwise, if the parameters affect the transition probabilities only, or the discount factor, the one-step reward might not be affected, but the solution of the MDP is.
>
> **Answer**: We do not put any restriction on the MDP parameter $\theta$, and the parameter can affect multiple things in the MDP problem, e.g., reward function and transition probabilities. This sentence [l.161] in the paper is to emphasize that the dependency of $\delta_\theta(\tau, \pi)$ on $\theta$ can come from the reward function $R_\theta$, where the dependency on $\theta$ can also come from other places, e.g., the expectation $\mathop{\mathbb{E}}\nolimits_{\mathcal{T} \sim \pi, \theta}$ over policy $\pi$ and transition probabilities dependent on $\theta$. In short, we do not require the reward to be the only dependency of the MDP parameter. We will update the draft to make this point clearer in the paper.
>
> **Question**: [l.258] I am not familiar with this snare-finding MDP setting, how is this not a bandit problem where each arm is the site to be visited and the reward is sampled with some probability? Here the reward doesn't depend on the set of features?
>
> **Answer**: This is not a multi-armed bandit problem because the snare can stay in the site if we do not remove it. Therefore, the snare-finding problem is essentially a sequential decision making problem where the state of the next time step is dependent on the state of the current time step.
>
> **Question**: [l.277] in this example, the reward doesn't depend on the set of features? Please state explicitly what is the set of features here.
>
> **Answer**: In the TB example, the reward does not depend on the missing parameter $\theta$ and feature $x$, but the cumulative return depends on $\theta$ and $x$ because the cumulative return depends on transition probabilities, which are a function of $\theta$ and $x$. We assume that there is a patient feature associated with each patient that we can use to infer the corresponding transition probability, e.g., the patients' historical adherence record.
>
>
>
> **Comment**: [Fig. 2] the color for the DF-PG-Identity setting needs to be changed, currently it's barely visible in the performance plots.
>
> **Answer**: We thank the reviewer for the feedback. We will change the plot colors in our final draft.

---

> > ### Comment · Reviewer_HiZT · 2021-08-23
> > **Response to Authors**
> >
> > Thank you for a detailed response. I would just like to add that it would be useful to make these things regarding the assumptions clear in the paper, this would improve readability a lot.
> >
> > > This is not a multi-armed bandit problem because the snare can stay in the site if we do not remove it. Therefore, the snare-finding problem is essentially a sequential decision making problem where the state of the next time step is dependent on the state of the current time step.
> >
> > Maybe you want to clarify this also in the main text, to make it explicit that it's not a bandit problem (although there is a reference).
> >
> > PS: good work

---

### Official Review · Reviewer_DDs9 · 2021-07-25

**Rating:** 8
**Confidence:** 4

**Summary:**

The authors extend the "predict-then-optimize" framework to sequential decision problems. The framework which has been studied previously focuses on improving the quality of the decision-making process compared to the traditional two-stage prediction-optimization approach. In this paper, the sequential decision-making problem is formulated as a MDP, with the goal being to learn a predictive model that can predict missing parameters. At training time, a predictive model to map features to missing parameters is learned, and at test time, the learned model is used to make predictions without using trajectories. The predicted parameters are used to solve the test MDPs, yielding an optimal policy which is evaluated using an offline off-policy evaluation.

The authors study two types of optimality conditions: (1) a policy gradient based approach where the expected cumulative reward is maximized, (2) a Bellman based optimality condition where mean squared Bellman error is minimized. These optimality conditions are expressed using KKT conditions. However, due to the large state and action spaces involved in the optimization reformulation, and the high-dimensionality of the policy space, the authors propose a technique to sample an estimate of the first and second order derivatives to approximate the optimality and KKT conditions. To handle the second challenge, two options are proposed - approximating the Hessian by constant identity matrix, and using a low-rank Hessian approximation and application of Woodbury matrix identity.

**Main Review:**

To the best of my knowledge, this is the first paper that extends the predict-then-optimize framework to sequential decision-making processes. The paper is well-written and easy to follow. The material in the main paper is mathematically sound. A few comments -

1. L227 - It is not clear to me why Eq(8) is "equivalent to the idea of differentiating through the final gradient of Bellman error..". Please elaborate.
2. L299-300 - While the experimental analysis as presented clearly shows that the proposed PG-W and Bellman-W outperform TS, it cannot be conclusively stated that they outperform the low accuracy approximation of the Hessian through a constant identity matrix. It is true in the case of the Snare dataset, but not the other two. Do the authors have a prescriptive guidance as to when the constant identity matrix based approximation may be preferable over the low-rank approximation. Additionally, since the latter is more expensive, is there a bound on how much worse will the former be than the latter?
3. Is it possible to extend Figures 2(c) and 3(c) to include backpropagation runtime for Bellman optimality condition?
4. The experimental results section could be further strengthened by benchmarking the proposed approach against comparable approaches - perhaps, Karkus et al. [15] and Futoma et al. [10].

**Time Spent Reviewing:**

5 hours

---

> ### Author Response · Authors · 2021-08-10
> **Connection to previous works + hypothesis about low-rank approximation v.s. constant identity matrix approximation + updated computation experiment**
>
> We thank the reviewer for the constructive feedback. We will add clarification to the paper to reflect the reviewer's comments. Please also find our answers to the reviewer's questions below.
>
> **Question 1**: [L227] It is not clear to me why Eq(8) is "equivalent to the idea of differentiating through the final gradient of Bellman error..". Please elaborate.
>
> **Answer**: If we use a negative identity matrix $cI$ with $c < 0$ to approximate the Hessian matrix $\nabla^2_\theta J_{\theta}(\pi^*)$ in Equation (8), the RHS of Equation (8) becomes:
> $$\frac{d ~ \text{Eval}(\pi^*)}{d w} \approx -c \frac{d \text{Eval}(\pi^*)}{d \pi^*} \nabla^2_{\theta \pi} J_{\theta}(\pi^*) \frac{d \theta}{d w} = -c \frac{d \text{Eval}(\pi^*)}{d \pi^*} \frac{\partial}{\partial w}(\nabla_{\pi} J_{\theta}(\pi^*))$$
> where the second equality is due to chain rule $\nabla^2_{\theta \pi} J_{\theta}(\pi^*) \frac{d \theta}{d w} = \frac{\partial}{\partial w} (\nabla_{\pi} J_{\theta}(\pi^*))$. The term $\frac{\partial}{\partial w} (\nabla_{\pi} J_{\theta}(\pi^*))$ is exactly the derivative of the gradient of Bellman error when using Bellman-based method, which matches the idea proposed by Futoma et al., differentiating through the last gradient of Bellman error to achieve end-to-end training. We will add this explanation to the paper.
>
>
> **Question 2**: [L299-300] Do the authors have a prescriptive guidance as to when the constant identity matrix based approximation may be preferable over the low-rank approximation
>
> **Answer**: As noted by the reviewer, sometimes using a constant identity matrix to approximate the Hessian can achieve better performance than using a more accurate low-rank approximation. We hypothesize that the use of less accurate Hessian approximation can help enlarge the variance of end-to-end gradient and thus improve the performance of gradient descent in non-convex settings by helping escape local minima. We can see that in the grid world example, the policy is solved by tabular Q learning and is thus more deterministic with less variance in either policy gradient or Bellman-based methods, where adding variance leads to better performance. On the other hand, in the snare finding and the Tuberculosis problem, the learned policy -- which is represented by deep neural networks -- is more stochastic with larger variance, where adding more variance may not help. This hypothesis suggests that the use of different approximations should depend on the stochasticity of the optimal policy.
>
>
> **Question 3**: Is it possible to extend Figures 2(c) and 3(c) to include backpropagation runtime for Bellman optimality condition?
>
> **Answer**: Please find the updated result of Figure 2(c) and Figure 3(c) in the following link:
>
> https://www.dropbox.com/s/jempv2kf2xm0xra/computation.pdf?dl=0
>
> We will update the plots in our final draft.
>
>
> **Question 4**: The experimental results section could be further strengthened by benchmarking the proposed approach against comparable approaches - perhaps, Karkus et al. [15] and Futoma et al. [10].
>
> **Answer**: The approach by Futoma et al. [10] reduces to our approach using Bellman-based optimality and an identity matrix to approximate the Hessian as explained in the previous question. On the other hand, the focus of Karkus et al. [15] is to learn and embed a representation to a robotic algorithm, where the algorithm is given and the algorithm is differentiable by unrolling directly. In this paper, we do not assume a given differentiable optimal policy/algorithm, where our focus is to make the optimal policy differentiable. The focus and the technical challenge involved in Karkus et al. [15] are different from ours. We thank the reviewer for the feedback, and we will add more explanation to the experiment section.
>
> [10] J. Futoma, M. C. Hughes, and F. Doshi-Velez.  Popcorn:  Partially observed prediction constrained reinforcement learning.arXiv preprint arXiv:2001.04032, 2020
>
> [15] P. Karkus, X. Ma, D. Hsu, L. P. Kaelbling, W. S. Lee, and T. Lozano-Pérez.  Differentiable algorithm networks for composable robot learning.arXiv preprint arXiv:1905.11602, 2019

---

> > ### Comment · Reviewer_DDs9 · 2021-09-01
> > **Feedback to authors**
> >
> > Thanks for your detailed response.
> >
> > Regarding your response to Q.4, are there perhaps other works that you could use for benchmarking?

---

> > > ### Author Response · Authors · 2021-09-03
> > > **Response to benchmark feedback**
> > >
> > > We are sincerely grateful to the reviewer for providing valuable feedback.
> > >
> > > In the literature of optimization problems using the predict-then-optimize framework, to our knowledge, unrolling (Stoyanov et al. 2011; Domke 2012; Amos et al. 2017) is considered the most common alternative of achieving end-to-end differentiability other than using KKT conditions. However, unrolling immediately becomes infeasible when the optimization problem complexity grows, where sequential decision-making deteriorates this scalability challenge. On the other hand, it is still possible to use unrolling to benchmark small sequential decision problems. Unrolling provides an expensive but accurate approach for comparison. We thank the reviewer for the constructive feedback. We will update the draft to reflect the reviewer's suggestion.
> > >
> > > Stoyanov, V., Ropson, A., & Eisner, J. (2011, June). Empirical risk minimization of graphical model parameters given approximate inference, decoding, and model structure. In Proceedings of the Fourteenth International Conference on Artificial Intelligence and Statistics (pp. 725-733). JMLR Workshop and Conference Proceedings.
> > >
> > > Domke, J. (2012, March). Generic methods for optimization-based modeling. In Artificial Intelligence and Statistics (pp. 318-326). PMLR.
> > >
> > > Amos, B., Xu, L., & Kolter, J. Z. (2017, July). Input convex neural networks. In International Conference on Machine Learning (pp. 146-155). PMLR.

---

### Decision · Program_Chairs · 2021-09-27

**Decision:**

Accept (Spotlight)

**Comment:**

The authors propose a "predict then optimize" approach for sequential decision problems. In other words, they consider a family of problems where features of the problem are given that are indicative of the MDP definition (reward and/or transition function). The proposed solution strategy is then a pipeline, where first missing MDP parameters are predicted and subsequently the MDP is solved using policy-based or value-based methods. Importantly, unlike e.g. many model-based RL approaches, this is not done in a 2-stage approach (where the dynamics model or reward function are trained on a separate supervised prediction loss) but end-to-end (the prediction of MDP parameters is learned to optimize the final RL loss). The main challenge in applying this framework is the large state-action spaces and large policy spaces concerned. So, the authors focus their attention on approximating key steps of the algorithm to obtain practicable algorithms in this setting.

The reviewers are unanimously positive about this paper.
- All reviewers consider the paper novel (though one reviewer considers it not entirely original)
- The reviewers consider the paper mathematically sound and the design of algorithmic components to be well justified
- The reviewers considered the paper to be well-written
- While the experiments to showcase clear benefits, they are mostly limited to toy tasks and the results could benefit from additional comparison
- Three reviewers did mention that the problem statement is not well-motivated. What in what 'real' settings can we apply this method?

Considering the unanimous positivity and agreement on novelty, soundness, and clarity, I recommend this paper to be accepted.